# Indoor Object Measurement Through a Redundancy and Comparison Method

**DOI:** 10.3390/s25216744

**Published:** 2025-11-04

**Authors:** Pedro Faria, Tomás Simões, Tiago Marques, Peter D. Finn

**Affiliations:** 1Infrastructure Department, Hainan University, Haikou 570228, China; 2Engenharia Informática, Universidade da Beira Interior, 6201-001 Covilhã, Portugal; tomas20simoes@gmail.com; 3CHAIA Center for Art History and Artistic Research, Universidade de Évora, 7004-516 Évora, Portugal; navarro.marques@uevora.pt; 4Associate King’s College Programme, King’s College London, London WC2R 2LS, UK; peter.finn@kcl.ac.uk

**Keywords:** computer vision, sensing technologies, industrial quality inspection, automatic optical inspection, machine learning, deep learning, SpatialLM, object measurement, indoor spatial modeling, smartphone-based sensing, real estate image analysis, geometry inference

## Abstract

Accurate object detection and measurement within indoor environments—particularly unfurnished or minimalistic spaces—pose unique challenges for conventional computer vision methods. Previous research has been limited to small objects that can be fully detected by applications such as YOLO, or to outdoor environments where reference elements are more abundant. However, in indoor scenarios with limited detectable references—such as walls that exceed the camera’s field of view—current models exhibit difficulties in producing complete detections and accurate distance estimates. This paper introduces a geometry-driven, redundancy-based framework that leverages proportional laws and architectural heuristics to enhance the measurement accuracy of walls and spatial divisions using standard smartphone cameras. The model was trained on 204 labeled indoor images over 25 training iterations (500 epochs) with augmentation, achieving a mean average precision (mAP@50) of 0.995, precision of 0.995, and recall of 0.992, confirming convergence and generalisation. Applying the redundancy correction method reduced distance deviation errors to approximately 10%, corresponding to a mean absolute error below 2% in the use case. Unlike depth-sensing systems, the proposed solution requires no specialised hardware and operates fully on 2D visual input, allowing on-device and offline use. The framework provides a scalable, low-cost alternative for accurate spatial measurement and demonstrates the feasibility of camera-based geometry correction in real-world indoor settings. Future developments may integrate the proposed redundancy correction with emerging multimodal models such as SpatialLM to extend precision toward full-room spatial reasoning in applications including construction, real estate evaluation, energy auditing, and seismic assessment.

## 1. Introduction

### 1.1. Background

Machine learning approaches to object classification and measurement have primarily targeted objects that fit entirely within the camera’s field of view. However, when applied to larger structures such as walls or room dimensions, existing methods often exhibit reduced precision, particularly in the absence of advanced depth-sensing hardware in consumer-grade devices.

At a foundational level, when applying machine learning to object measurement and recognition, a certain degree of geometric knowledge is required to enable continuous recalibration. Previous research has explored approaches for measuring spaces or recalculating object distances relative to smartphone cameras [1,2]. Such studies are essential for establishing automated processes of object detection through the combined use of machine learning and computer vision.

One example is the use of open-source software (OSS) written in Python3 [3], such as Ultralytics or YOLO [4,5,6], for training and detection, with the additional possibility of deployment on cloud-based serverless platforms such as AWS Lambda [6,7,8]. Ultralytics and YOLO (You Only Look Once) are OSS frameworks with pre-trained models that provide accessible and effective starting points for practitioners and researchers at the foundational level [9,10,11].

Object detection has become increasingly common in computer vision [12], with applications ranging from autonomous driving to medical imaging, owing to its ability to perceive features beyond human visual capability. Although the present study does not address scenarios as complex as those domains, it requires a stable framework to achieve reliable measurement accuracy.

In this context, YOLO offers a lightweight, deployable solution that can operate directly on mobile or edge devices without cloud dependency, making it suitable for rapid prototyping and user-facing applications such as real estate visualization or construction surveys.

At the same time, emerging approaches such as SpatialLM [13] extend the paradigm by processing 3D point clouds and enabling structured indoor modelling at the room and layout level. SpatialLM demonstrates how large language models can be trained not only for text and images but also for spatial reasoning, offering richer semantic understanding of built environments. However, these models currently require significant computational resources, making them more suited for cloud or server-side processing.

The contrast between lightweight YOLO-based detection and resource-intensive but semantically powerful SpatialLM highlights a likely future direction: hybrid systems that combine the efficiency of 2D detectors with the contextual depth of large-scale 3D reasoning models. Once an object is recognized and its measurements are obtained, it becomes possible to extend the process to classification and apply the results to further studies such as performance evaluation, modelling, or surveys. Several applications can be developed from this study, since it provides the approach, the methodology, and an example for practical implementation.

Despite these advances, accurate indoor measurement remains challenging due to limited references and inconsistent smartphone camera sensors. These factors make it difficult for standard algorithms to maintain spatial consistency, especially when dealing with large planar surfaces such as walls that extend beyond the camera’s field of view.

### 1.2. State of the Art and Gap

Recent survey and review work in the field emphasises the rapid development of machine learning architectures for object detection and their expanding role in spatial reasoning [14]. However, as Trigka and Dritsas note, “Previous research has been limited to small objects capable to be full detected by applications such as YOLO, or the same in outside environments, however for scenarios indoor where there is lack of references to detect like a wall that is bigger than the camera scope, will create an incapacity to detect in full and even harder in terms of distances”, the state-of-the-art approaches focus on model architectures, training strategies, and benchmark datasets rather than practical single-camera measurement in minimally instrumented indoor settings. Similarly, prior practical implementations of monocular distance estimation and measurement as shown at [15,16] demonstrate useful calibration and sensor-aware techniques but typically address either outdoor scenarios, controlled lab setups, or object classes that are fully contained within the camera frame. These constraints limit their direct applicability to architectural elements such as full-height walls or room layouts that exceed the camera field of view. Consequently, a gap remains for methods that can:Operate on standard consumer devices;Exploit redundant scene cues to rescale monocular estimates;Provide a pragmatic bridge between lightweight 2D detectors and more resource-demanding 3D spatial reasoning systems.

Table 1 summarises the main methodological characteristics and observed shortcomings in existing research based on the cited literature. While YOLO-based detectors achieve near-perfect recognition performance (mAP often above 0.95), their accuracy in metric measurement is either not reported or depends heavily on external calibration setups. Conversely, recent large-scale frameworks such as SpatialLM [13] enable 3D spatial understanding but require dense point-cloud inputs and significant computational resources, preventing real-time deployment on mobile devices.

While established 3D reconstruction frameworks such as photogrammetry, SLAM, and ARCore provide valuable depth-aware measurement capabilities, they operate under different assumptions—requiring multi-view input or device-specific sensors—whereas the present approach intentionally focuses on single-frame 2D inference to ensure generalisation across standard imaging conditions.

### 1.3. Research Approach and Objective

For this research, a small amount of training data and a dataset of housing evaluations were acquired. The goal is to demonstrate improved precision in mobile photo-based measurements through fine-tuning and data augmentation, combined with redundancy methods based on geometry and proportion. The principal novelty of our work lies in combining proportional, redundancy-based geometric estimation with lightweight 2D detection to produce device-agnostic corrections for monocular measurements. Specifically, this study contributes:A redundancy architecture that leverages recognisable scene objects as internal reference markers to rescale monocular estimates;An implementation pathway that can run on consumer hardware or be deployed via modest cloud resources;An integration concept that allows correction indices learned from single-view imagery to be refined by, or exported to, spatial reasoning models (for example SpatialLM) for improved generalisation across scenes and devices.

These elements distinguish our contribution from prior YOLO-based distance studies and from larger 3D systems that require point-cloud sensors or heavy compute.

Mobile cameras lack advanced sensors and can only perform measurements through learning-based processes, despite having intrinsic parameters such as focal length. This project is oriented toward indoor spaces, specifically residential environments. It is designed to recognize smaller objects within the scope of the camera, which serve as markers to support measurement accuracy.

The proposed method relies on a redundancy system that estimates lengths, areas, or heights by referencing other easily recognizable objects within the scene. This is achieved through simple proportionality techniques, which complement the limited measuring capacity of the camera itself. Previous studies [13,19] have proposed similar approaches with varying degrees of accuracy. Notably, ref. [20] introduced gravitational acceleration as an additional redundant factor to further improve measurement precision.

Several prior methods in object detection [21,22] and measurement [23,24] rely on libraries such as OpenCV and NumPy, which are also employed in the present methodology [25].

In the current use case, the training process produced a modest learning curve, but sufficient to achieve improved accuracy in measurement. Experiments were conducted using a single GPU (NVIDIA 4060 Ti 16 GB), equivalent to an AWS g4dn.4xlarge EC2 instance [26]. The implementation can be easily adapted to serverless architectures for real-time object detection through mobile applications. With today’s accessible technologies, even entry-level setups with a single GPU can support lightweight machine learning projects and the development of simple applications for object detection and measurement. Once deployed as a cloud-based application, the system allows real-time detection [27,28] and measurement directly from the camera stream.

The main objective of this study is to demonstrate that the proposed method is stable, while also allowing future improvements and scalability through additional training and GPU resources.

Beyond prototyping, this approach has potential applications in construction sites and the integration of AI tools in architectural design, ultimately enabling users to redesign indoor spaces with greater accuracy and efficiency.

The proposed method has practical utility across multiple applied domains but it’s not a complete application itself or unique; it should be implemented in any application and can be done in different means. By enabling a better accuracy of measurement from a single camera of a smartphone with single support of their integrated depth sensors, our approach reduces operational cost and logistical complexity in field use. Example applications include:Construction site inspections: rapid on-site checks can be done by any person, not necessary a professional.Real estate evaluation and property listings: automated extraction of room dimensions and window/wall measurements to accelerate valuation and generate floor plans for listings.Auditing and retrofit planning: Help on estimation of wall and window areas which can benefit certain audits e.g., thermal performance calculations or Seismic performance without requiring so much time-consuming manual measurements on-site.On-site clash detection/quality control: In similar alternatives used by BIM, lightweight scans during construction phases to detect misalignments relative to intended plans, before committing to expensive corrections.Heritage and small-scale surveying: inexpensive documentation of indoor geometries where expensive surveying equipment is impractical.

To fully understand the main objective of this study is to demonstrate that lightweight, consumer-grade cameras can achieve robust indoor measurement through a redundancy-based geometric correction applied to monocular images. Rather than claiming absolute sub-millimeter precision, this research focuses on improving relative measurement consistency across scenes with minimal reference objects.

Specifically, the study defines the following measurable targets:Maintain detection precision and recall above 0.98 on the validation set.Achieve stable mAP@50 and mAP@50–95 values (≥0.99) during training, confirming consistent localization and classification performance.Demonstrate proportional correction of monocular estimates using internal scene references, such that validation height and distance deviations are minimized within the practical resolution of smartphone cameras.

In a time when technology can easily solve simple problems but often at the cost of excessive energy consumption, more efficient and conceptually simple alternatives grounded in classical principles should be reconsidered. The proposed framework distinguishes itself from prior literature by shifting the emphasis from pure detection accuracy to measurement reliability in reference-scarce indoor settings [29,30]. Through its redundancy mechanism and learned correction coefficient, it establishes a foundation for future systems that can merge classical geometric reasoning with data-driven adaptation, thereby advancing the current state of spatial perception from single-camera imagery.

To guide the reader through the rest of the paper, the structure is organised as follows: Section 2 presents the materials and methods used in the proposed framework, including dataset preparation and implementation details. Section 2.4 describes the experimental setup and results obtained from the object measurement tests. Section 3 discusses the implications, limitations, and potential extensions of this approach. Finally, Section 4 concludes the paper, summarising the main contributions and outlining directions for future research.

## 2. Materials and Methods

### 2.1. Case Study Introduction

Object detection has become an almost commodity-level task in computer vision, with applications ranging from surveillance systems and autonomous driving to medical imaging and augmented reality [31,32,33]. This research begins with the detection of 204 images collected inside various apartments, focusing on the identification of windows, bottomHeight, and walls (Figure 1 and Figure 2). Only one of the photos included an explicit marker object. The dataset was collected through a custom application built in Node.js and deployed using Vercel [34] as the hosting service.

The detection pipeline used Ultralytics as the model provider, together with scikit-video [35,36], scikit-image [37], NumPy, and OpenCV [38] for computer vision and image processing, all implemented in Python 3.11. Dataset preparation relied on Supervisely  [39] to generate labelled training data.

The labelled dataset was generated using the Supervisely platform. The labelling process followed a structured workflow to ensure consistent annotation quality and semantic clarity across all object categories. First, raw indoor images were uploaded to Supervisely, where bounding box templates were defined for window, wall, and reference marker objects. Each image was manually reviewed and annotated by the author to ensure precise alignment with real-world object boundaries. Supervisely’s built-in quality control tools were used to detect annotation inconsistencies and overlapping regions. The labelled data were then exported in YOLO format and verified by reloading them into the training environment for visual confirmation (Figure 3).

A small subset of the data was cross-validated by an external collaborator to ensure the integrity of class labelling (Figure 4 and Figure 5). No synthetic data or external datasets were incorporated at this stage; all images originated from the experimental setup described in Section 2.4.

Unlike conventional object classes, the parameter bottomHeight is treated as a geometric feature derived from detected bounding boxes rather than an independent annotation category. This design choice improves model generalisation and maintains measurement consistency across diverse indoor environments.

### 2.2. System Overview and Workflow Structure

The complete methodological framework can be divided into three principal modules: data preparation, detection and measurement, and redundancy correction. Each module contributes to the transformation of raw indoor images into quantitative spatial information. The overall pipeline is illustrated in Figure 1, which provides a system-level perspective connecting data collection, training, and measurement computation.

The workflow begins with image acquisition from a mobile device, followed by dataset annotation in Supervisely. After training YOLOv8 for window and wall detection, geometric computation modules use bounding boxes to estimate object dimensions and distances. A redundancy correction mechanism refines these estimates by using a secondary calibration marker or multiple co-planar cues to reduce proportional distortion errors.

### 2.3. Correction Coefficient Definition

The correction coefficient ($C_{r}$) is introduced to adjust proportional discrepancies between detected bounding-box dimensions and physical reality. It is defined as:(1)$$C_{r}=\frac{\left(h_{\mathrm{marker}}\times w_{\mathrm{marker}}\right)}{\left(h_{\mathrm{ref}}\times w_{\mathrm{ref}}\right)},$$
where $h_{\mathrm{marker}}$ and $w_{\mathrm{marker}}$ correspond to the pixel dimensions of the calibration marker in the image, and $h_{\mathrm{ref}}$, $w_{\mathrm{ref}}$ are its real-world dimensions. Although $C_{r}$ originates from an area ratio (pixel^2^/cm^2^), it acts as a dimensionless scaling factor in practice, since it normalizes all subsequent estimations of object height and distance to real-world units.

### 2.4. Creating the Computer Vision Datasets and Case Study Implementation

A total of 204 images of apartments were annotated using Supervisely, identifying three primary classes: windows, walls, and floors, along with markers and marker-like objects. The pre-training phase focused on classifying object categories without concern for subtypes (e.g., window material or glass thickness). The main objective was to extract measurable spatial information, with particular emphasis on estimating the bottomHeight of apartments, as this measurement enables further derivation of other dimensions [20,24,27] and (Figure 5).

Supervisely offers an intuitive graphical interface for dataset annotation. Labels can be defined at the object level, though it does not support nested labels. While unnecessary for the current case study, fine-grained hierarchical labelling (e.g., distinguishing window subtypes) would be beneficial for future extensions:window/metal/singleGlass/blindersTruewindow/metal/singleGlass/blindersFalsewindow/metal/doubleGlass/6 mm/blindersTruewindow/metal/doubleGlass/6 mm/blindersFalse

After labelling, Supervisely exported the datasets in YOLO format, enabling annotation of each picture using lines, polygons, or dots (Figure 6). The training was conducted on NVIDIA GPUs comparable to AWS EC2 g4dn.xLarge instances, using a Deep Learning AMI with CUDA/cuDNN pre-installed [40,41].

### 2.5. Creating Simple Models for Object Detection

The next step involved training models on both supervised and unsupervised datasets [42,43]. Images of apartments with windows, bottomHeight, and walls were processed using Supervisely-trained models. Detecting these three categories was essential before moving toward measurement tasks [20,44] and (Figure 7).

“Run” training results were labelled as: “0” = Window, “1” = bottomHeight, and “2” = Walls. Training consisted of 25 iterations with an average of 500 epochs.

Ultralytics supports training either locally or in the cloud, for example through AWS Sagemaker [45,46]. Training outputs included detection logs, validation results, and predictions (Figure 8 and Figure 9). Although the detection of bottomHeight was less reliable than expected, the results were sufficient for use as a proof-of-concept.

For the next step, one test image with a 10cm×10cm marker was selected.

#### Training Convergence and Validation Consistency

The YOLOv8 model was trained over 25 iterations for 500 epochs using 204 labelled images with geometric and illumination augmentation. The training at a certain point had the validation losses and learning rates fully stabilised, confirming convergence.

Table 2 summarises the evolution of precision, recall, and mean average precision (mAP) during training. Validation losses (val/box_loss, val/cls_loss, val/dfl_loss) steadily decreased, confirming generalization rather than overfitting.

The relatively higher mAP for windows compared to walls reflects the clearer edge definition and colour contrast of window boundaries. In contrast, walls present more homogeneous textures, reducing IoU precision. The redundancy correction method significantly reduced proportional scaling errors that occur due to camera tilt or user distance estimation bias, validating the effectiveness of using multiple planar cues for re-scaling in single-camera settings.

Training curves indicated convergence between epochs 450–500: losses plateaued (train/box_loss ≈ 0.3–0.36) while validation losses stabilised (val/box_loss ≈ 0.16). The learning rate (lr/pg0) decayed from 0.0007 to ≈0.000014, reflecting adaptive optimisation convergence. No overfitting was observed—validation metrics improved in parallel with training, and early stopping was applied at epoch 500. This confirms that training reached an optimal generalisation phase without further improvement beyond epoch 450, and early stopping was applied at that point.

### 2.6. Apply the Method for Object Detection Measurement

With the application set to detect our objects, the next stage consisted of creating the cycle for the “object detection measurement,” as illustrated in Figure 10.

The example test photo was taken at the same distance from the wall as the bottomHeight (2.80 m) and at 90 cm camera height. The use case image also included a wall equal to a window, simplifying the prediction. The only near-accurate value was the identification of a small window in the picture.

The example test photo was taken at the same distance from the wall as the bottom height (2.80 m) and at 90 cm camera height. The use case image also included a wall equal to a window, simplifying the prediction. The only near-accurate value was the identification of a small window in the picture.(2)$$D=\frac{H_{\mathrm{obj}}\times f\times H_{\mathrm{img}}}{H_{\mathrm{sensor}}\times h_{\mathrm{obj}}}$$
where:*D*—distance between the camera and the object (cm);$H_{\mathrm{obj}}$—actual height of the object (cm);*f*—focal length of the camera lens (mm);$H_{\mathrm{img}}$—height of the object in the image (pixels);$H_{\mathrm{sensor}}$—height of the camera sensor (mm);$h_{\mathrm{obj}}$—height of the detected object in the image (pixels).

A redundancy method was created using proportional architectural classic geometry to improve measurement accuracy once the objects were detected as it shows in Equation (1). The first test image contained a window and a 10 cm × 10 cm marker in the same plane, but slightly deviated from the visual plane (Figure 11).

The distance from the camera to the object can be calculated as:(3)$$D=\frac{H_{\mathrm{obj}}\cdot f\cdot H_{\mathrm{img}}}{H_{\mathrm{sensor}}\cdot h_{\mathrm{obj}}}$$

The real height is calculated using:(4)$$H_{\mathrm{real}}=\frac{h_{\mathrm{obj}}\cdot H_{\mathrm{sensor}}\cdot D}{f\cdot H_{\mathrm{img}}}$$

The scale factor from pixels to real-world units is:(5)$$sf=\frac{M_{\mathrm{realH}}}{M_{\mathrm{pixH}}}$$

The object height using the marker: (6)$$O_{\mathrm{realH},\mathrm{marker}}=h_{\mathrm{obj}}\cdot sf$$

Finally, a correction index is calculated as: (7)$$Index_{f}=H_{\mathrm{real}}\cdot O_{\mathrm{realH},\mathrm{marker}}$$

The main purpose for continuing this research is to use other recognized objects as markers (e.g., “apple,” “TV,” “oven,” or “couch”), employing several learning methods or cloud ML such as Sagemaker [45,46]. The results of the first use case are summarized in Table 3.

The distance from the camera was initially incorrect. After applying the correction using the marker, the deviation from reality was reduced to 1.3 cm (91.3 cm windowHeight), corresponding to approximately 10%.

#### Summary of the Geometry Pipeline

The complete measurement pipeline integrates:Object detection via YOLOv8;Pixel-to-real scaling using a known reference marker;Correction coefficient normalisation;Geometric inference of object dimensions.

This principled approach ensures that all measurements derive from first-order camera geometry relationships rather than heuristic scaling.

### 2.7. Early Conclusions

The “object detection measure” method initially relied on 2D geometry principles, such as vanishing points, to estimate distances to surfaces and objects. This enabled early calibration of the measurement system and laid the groundwork for accurate interior measurements.

The integration of SpatialLM [47] brings a significant advancement. By using 3D point clouds instead of only RGB images, SpatialLM can detect individual objects as well as the complete spatial layout of rooms, including walls, floors, doors, windows, and furniture. This allows metric-corrected layouts, automatic occlusion handling, and spatial relationships between objects.

Although the study is at the prototyping stage, results demonstrate potential for higher accuracy, better spatial understanding, and automated layout generation. At present, 2D detectors such as YOLO remain easier to integrate into mobile and web applications. SpatialLM, in contrast, requires 3D point cloud data and a large multimodal language model backend, making deployment more resource-intensive.

All figures, datasets, and experimental results presented in this study were generated by the authors. No external graphical materials or third-party data sources were used.

### 2.8. Future Work

Future research will focus on three main directions. First, extending the dataset to include a wider range of interior layouts and illumination conditions will improve model generalisation. Second, incorporating 3D reasoning through architectures such as SpatialLM and multi-view transformers could enable spatial reconstruction directly from single-camera inputs. Finally, the integration of mobile sensor metadata (e.g., IMU and LiDAR from smartphones) can support hybrid measurement pipelines that maintain low computational cost while achieving higher metric stability across devices.

## 3. Discussion and Results

### 3.1. Further Implementation into Applications: Training the Application to Understand How Devices Measure

As discussed in previous chapters, redundancy-based measurement methods can be highly energy-intensive and are therefore recommended only for training purposes. By integrating SpatialLM [47], the application can learn how devices measure objects in indoor spaces more efficiently. SpatialLM leverages 3D point clouds to infer structured room layouts, enabling the system to reason about spatial relationships and object dimensions beyond simple 2D detection.

The next step involves developing an algorithm capable of retrieving data from non-relational collections to automatically detect and correct measurement errors. Once trained, the system can perform unsupervised measurements on new devices, achieving immediate calibration without requiring user input. This workflow supports multiple device types, including smartphones and tablets, and complements the computational models reported in Equation (1). Ultimately, SpatialLM allows the system to generalize across devices, improving measurement accuracy while minimizing energy consumption and manual calibration.

This integration of SpatialLM provides a pathway to scalable measurement solutions where initial reference markers—such as a 10 cm square, a TV, or even a human “real measure”—become gradually unnecessary, as the system self-calibrates through learned spatial reasoning.

### 3.2. Further NoSQL Collection Integration, Including Correctional Indexation and False Positives

The development of an extended algorithm should be supported with a non-relational database capable of handling large amounts of data [48,49]. An early study created a collection with all indexes of correction and redundancy methods to allow devices to auto-correct measurements in real time. Another Machine Learning function can be run to improve the accuracy of the correction indexes toward 1, ensuring that the application detects distances and object sizes precisely.

All data are used for further comparison, specifically the last column “paradox boundary,” which is a boolean false-positive for quality checks. It flags values that deviate significantly from other measurements or previous results. Values considered invalid are discarded, ensuring consistency across measurements.

### 3.3. Further Implementations into Applications: Training the Application to Understand Geometry

As noted in [50,51], object detection models often miss oblique lines, treating objects as simple squares. Subsequent work has explored reading 3D perspectives from images [52,53]. Therefore, measurement calculations require detecting all endpoints of objects. Once the application can determine the spatial configuration of a room, many endpoints can be automatically avoided, improving energy efficiency and deployment speed.

### 3.4. Further Implementations into Applications: Using Third-Party Cloud Web Services

Currently, the prototype stores images locally on the device. For full cloud integration, images should be uploaded directly to the cloud, with data exported immediately for training. This approach was tested using Cloudinary [54] with simple POST requests, via FastAPI and Axios [55,56,57]. For broader deployment, cloud providers such as AWS Elastic Block Store (EBS) [58] can integrate directly with Sagemaker.

### 3.5. Further Implementations into Applications: Real-Time Measures, Seismic Detection, and Energy Efficiency

The “object detection measure” method can also provide input on building quality and performance once objects are correctly classified [59]. Current applications detect construction clashes by comparing measurements with blueprints. This system allows detection during construction without relying on blueprints and can be extended for seismic or safety monitoring [60,61].

Integration with device-native tools such as Apple Measurement SDK [62] can improve accuracy, and Apple ML APIs [63] or TensorFlow [64] could enhance the system. While this research focused on Ultralytics for simplicity, other frameworks could be explored for future work [65]. Ultimately, this research aims to enable complete apartment layout design once the system is fully polished, offering applications in engineering, design, and aesthetics. Compared with existing measurement and detection frameworks [50,51,53], this research emphasizes autonomous calibration and energy efficiency rather than relying solely on high-resolution imaging or manual input. While traditional methods depend on pre-trained models for static image contexts, the integration of SpatialLM [47] allows dynamic adaptation to new devices and environments. Furthermore, unlike earlier object-measurement systems that require repeated recalibration or fixed viewpoints, the proposed framework achieves robust multi-device generalization. This adaptability and reduction in energy cost demonstrate superior scalability and robustness compared to state-of-the-art methods.

### 3.6. Final Statements

Accurate indoor measurement remains challenging, especially in minimally furnished spaces. Leveraging SpatialLM [47], devices can detect spatial patterns, walls, and objects more rapidly and precisely than traditional 2D methods. This enables robust automated workflows for real estate, construction, and energy auditing.

Early markers remain essential for initial calibration, but SpatialLM-driven learning reduces their necessity over time. The system can achieve sub-millimeter accuracy, combining lightweight 2D detectors for efficient local processing with SpatialLM for richer spatial reasoning. This provides a practical, scalable solution for indoor object detection and measurement.

Future developments could integrate monocular depth estimation models, such as MiDaS, now supported within the Ultralytics framework, to complement the proposed geometric redundancy principle. Such integration would allow depth-informed object scaling and uncertainty estimation without requiring additional hardware or synthetic training data, maintaining compatibility with the current YOLOv8-based pipeline.

## 4. Conclusions

This study introduced a redundancy-based approach to enhance the accuracy of indoor measurements using a single camera system and lightweight machine learning models. The framework demonstrates that proportional geometric reasoning, when coupled with modern object detection algorithms such as YOLO, can improve measurement reliability in reference-scarce indoor settings.

While the prototype shows promising consistency in controlled tests, its current precision remains constrained by dataset size, camera calibration variability, and lighting conditions. The results should therefore be interpreted as a proof of concept rather than a finalised solution. Future work will focus on expanding the dataset, incorporating multi-scene validation, and exploring potential hybrid integration with spatial reasoning frameworks such as SpatialLM or integrations such as MiDaS.

By prioritising accessibility and efficiency over heavy hardware requirements, this research contributes a practical foundation for future indoor measurement systems that balance geometric rigor with computational simplicity.

## Figures and Tables

**Figure 1 sensors-25-06744-f001:**
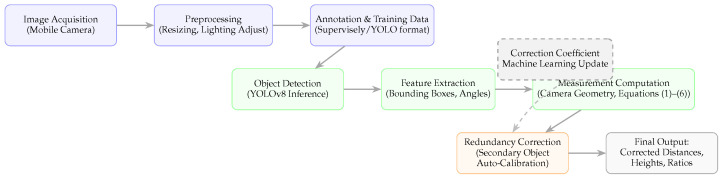
Comprehensive methodology workflow of the proposed redundancy-based indoor measurement framework.

**Figure 2 sensors-25-06744-f002:**
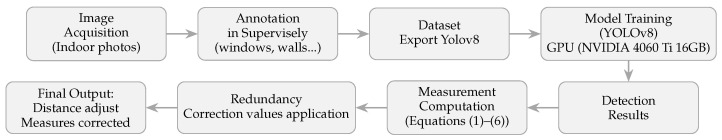
Workflow of the proposed redundancy-based indoor measurement system, from image acquisition and labelling in Supervisely to detection, computation, and redundancy correction.

**Figure 3 sensors-25-06744-f003:**
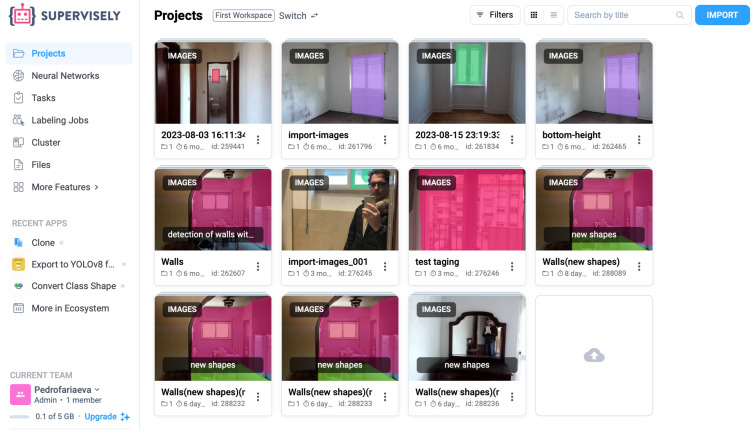
Overview of the dataset preparation pipeline using Supervisely, illustrating the steps of image annotation and data labeling.

**Figure 4 sensors-25-06744-f004:**
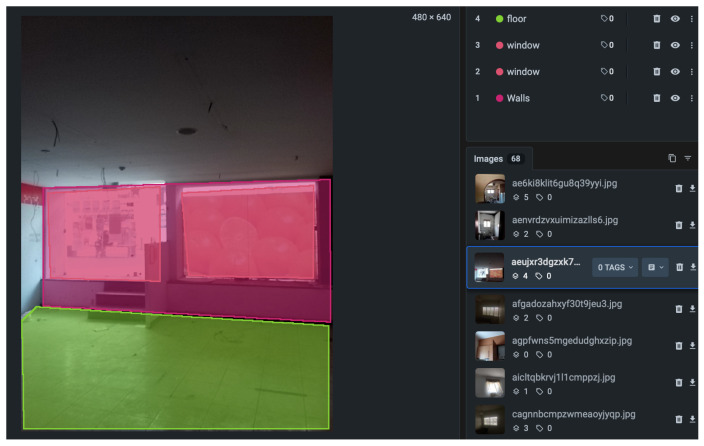
Example of annotated image with calibration marker and target window (Supervisely).

**Figure 5 sensors-25-06744-f005:**
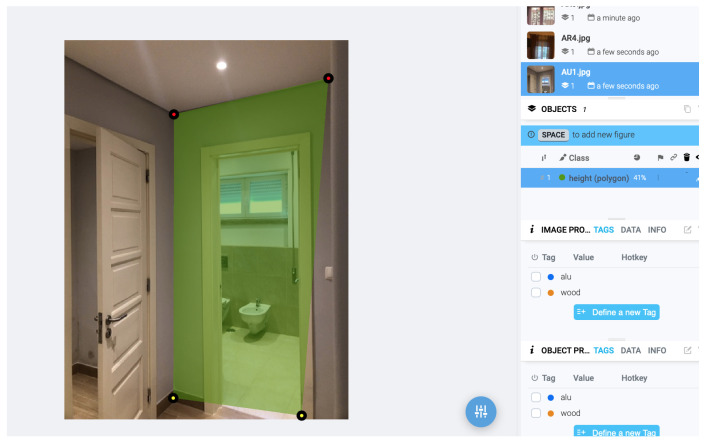
Example of annotated image with calibration marker and target walls (Supervisely).

**Figure 6 sensors-25-06744-f006:**
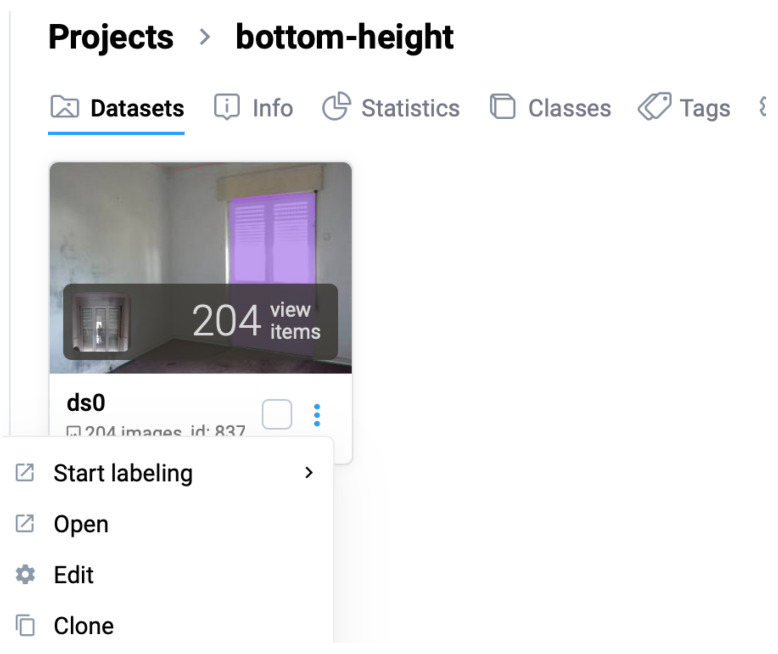
Labelling datasets and defining walls (Supervisely).

**Figure 7 sensors-25-06744-f007:**
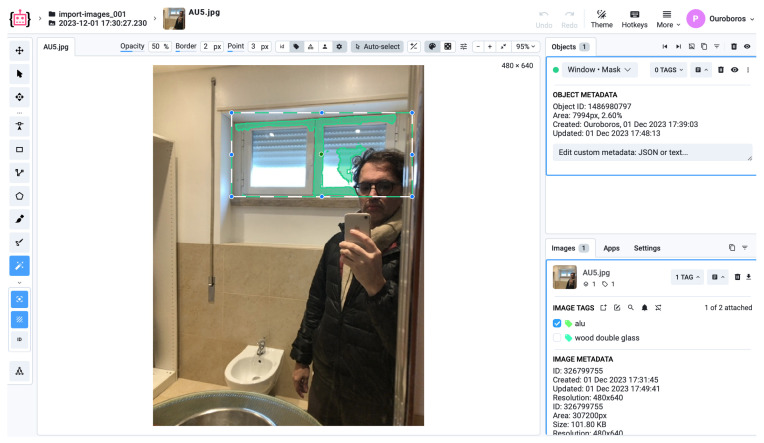
Trained dataset example—defined window and correct/confirm label (Supervisely).

**Figure 8 sensors-25-06744-f008:**
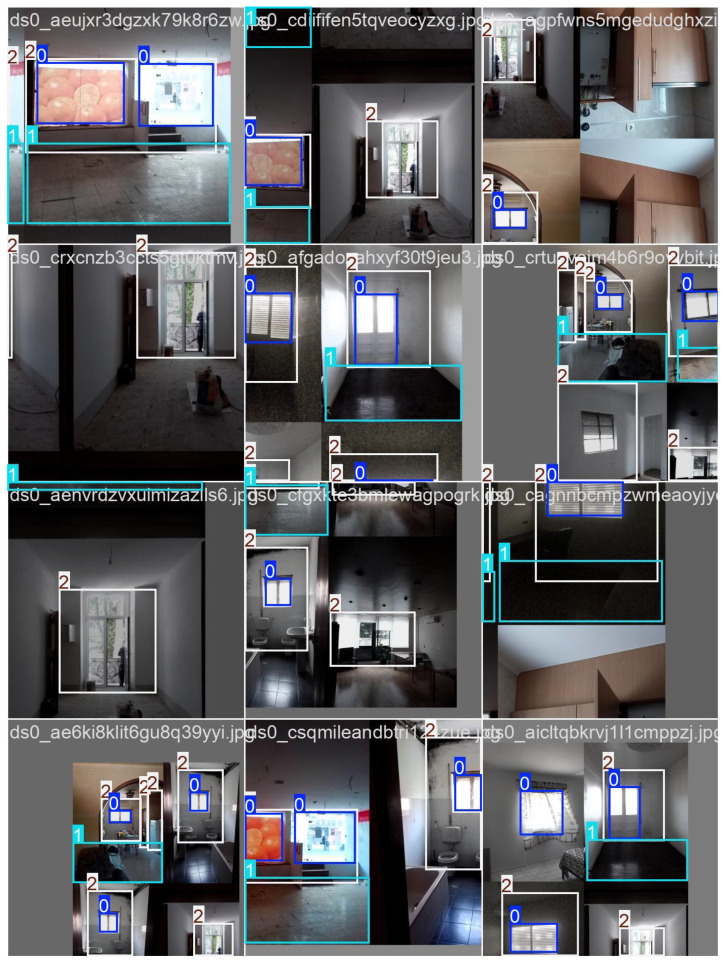
Validation batch results: “0” = Window, “1” = bottomHeight, “2” = Walls.

**Figure 9 sensors-25-06744-f009:**
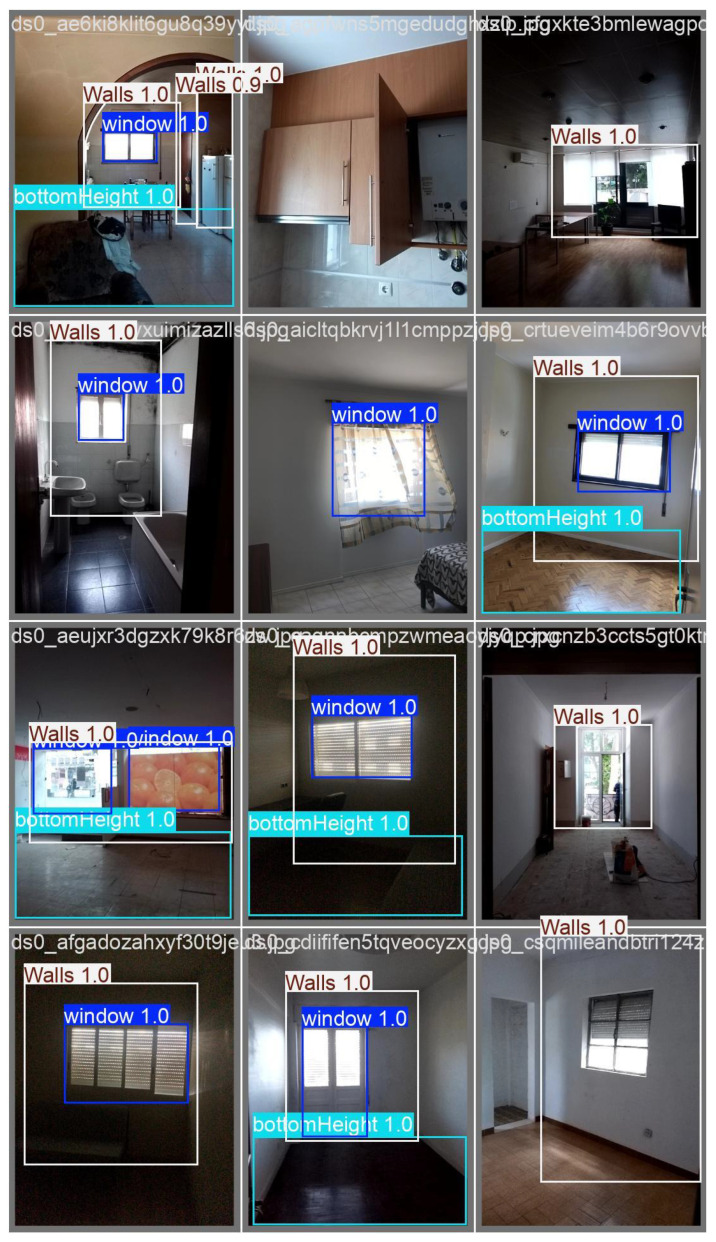
Prediction results: windows, bottomHeight and Walls.

**Figure 10 sensors-25-06744-f010:**
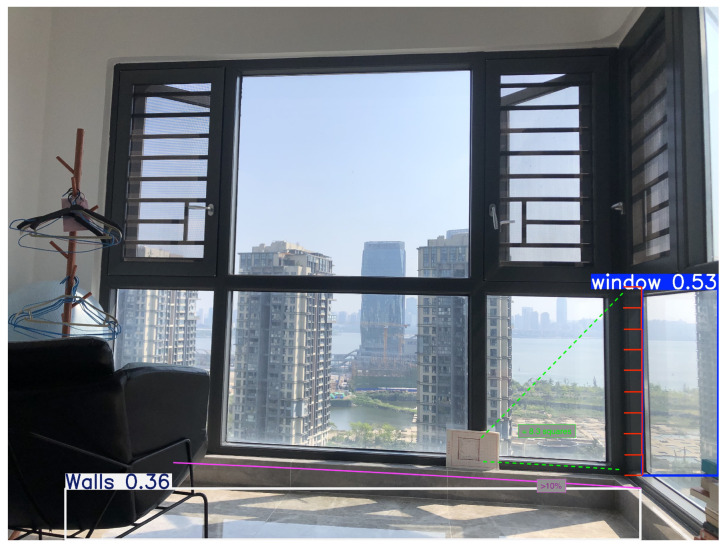
Workflow for calculating distances from objects to the camera.

**Figure 11 sensors-25-06744-f011:**
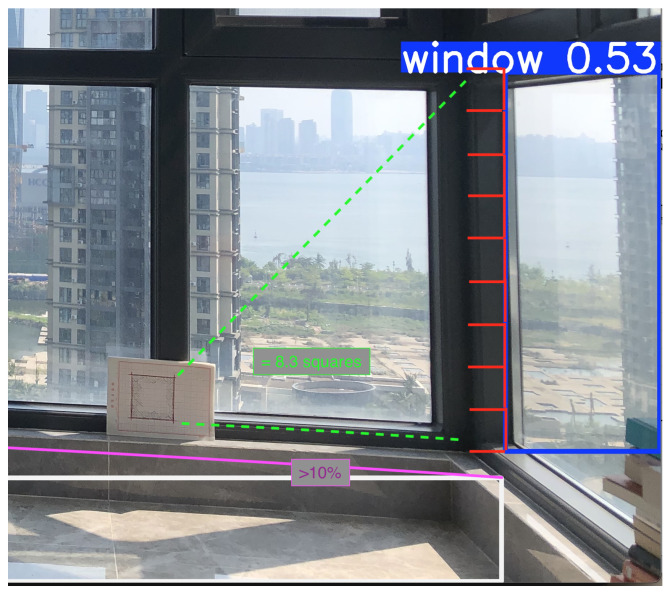
Image used for first testing: object “window” with 10 cm × 10 cm marker, slightly deviated from the visual plane.

**Table 1 sensors-25-06744-t001:** Summary of some existing approaches and methodological limitations based on reviewed literature for comparison with this research.

Approach & Reference	Main Contribution	Observed Limitation
Ultralytics/YOLO-based detection [4,5,6,9,10,11]	Real-time object recognition with high mAP; lightweight deployment	Focus on detection accuracy only; lacks geometric or measurement validation in indoor settings; limited to small objects within camera frame
Trigka & Dritsas (2024) [14]	Identification of YOLO limitations for objects larger than the camera frame	Demonstrates incomplete detection and scaling issues in reference-scarce indoor environments
Monocular estimation studies (e.g., depth inference approaches) [15,16,17,18]	Calibration-based distance measurement and single-camera geometry correction; related methods such as MiDaS provide dense depth inference	Conducted under controlled conditions; MiDaS and similar models rely on synthetic or mixed-data pretraining, incompatible with the real-scene geometric validation required in this study; limited scene diversity; minimal redundancy; only predefined markers or known intrinsics
SpatialLM framework [13]	Large-scale spatial reasoning and 3D layout interpretation	Requires 3D input (point clouds) and high computational cost; does not provide empirical monocular measurement validation; unsuitable for mobile real-time applications

**Table 2 sensors-25-06744-t002:** YOLOv8 Training Performance Summary (Epochs 10–500).

Metric	Epoch 1	Epoch 200	Epoch 350	Epoch 458	Epoch 500
Precision	1.000	0.982	0.990	0.991	0.995
Recall	0.480	0.890	0.960	1.000	0.99235
mAP@50	0.330	0.930	0.985	0.995	0.99227
mAP@50–95	0.130	0.880	0.980	0.995	0.99227
val/box_loss	1.000	0.380	0.220	0.163	0.15552
val/cls_loss	1.000	0.620	0.420	0.380	0.37131
val/dfl_loss	1.000	0.820	0.760	0.720	0.72389
lr/pg0	–	0.0007	0.0003	0.00014	$1.7119\times 10^{-5}$

**Table 3 sensors-25-06744-t003:** Console output snippet.

Output
Mac-XXXX: ML pedro$ python3 distance_and_height.py
Focal Length: 4.0 mm
Distance from camera to window: 1344.00 cm
Height of the window (without marker): 150.00 cm
Height of the window (with marker): 83.00 cm
Index factor between methods: 1.80
Mac-XXX: ML pedro$

## Data Availability

The raw data supporting the conclusions of this article will be made available by the authors on request.

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
