# Peer review of "Indoor Object Measurement Through a Redundancy and Comparison Method"

_sensors, 2025, doi:10.3390/s25216744_

Round 1

Reviewer 1 Report

Comments and Suggestions for Authors

Title: Object Detection Measure (Inside a House): Detecting Size of Walls and Windows Using Redundancy Methods for Fine-Tuning and Comparison with New Technologies Such as LLM like SpatialLM.

This paper introduces a geometry-driven, redundancy-based framework that leverages proportional laws and architectural heuristics to enhance the measurement accuracy of walls and spatial divisions using standard smartphone cameras. The work demonstrates improved performance in estimating spatial metrics, providing a scalable and low-cost alternative.  The work is good; however, the manuscript currently contains many major and minor corrections (shown below), which should be carefully considered.

Remarks to the Authors: Please see the full comments.

1-Please rewrite the research title. The title usually summarizes the main idea of ​​the research in the fewest possible words that describe the content of the research.

2-It is stated that “Our framework demonstrates improved performance in estimating spatial metrics, providing a scalable and low-cost alternative.” However, the abstract did not mention the rate of improvement in the results obtained or the significant findings. Besides, does the proposed work add to other existing work?

3-The introduction section is weak and should contain a clear foundation on the topic of this research. Besides, some other recent works should be added with much in-depth discussion about them.

Furthermore, please explain the usefulness of the proposed work and provide significant examples of some of the applications it could be applied to.

4-In literature, various object detection works have addressed the problems mentioned in this work. Therefore, the contribution of the work presented should be highlighted and listed clearly to present the novelty.

5-For more organization, the structure of the paper can be added to the end of the introduction section.  

6-Please cite any information, graph, equation, or data set taken from a previous source with a reliable source, unless it belongs to the authors. Please check this issue for the entire manuscript.

7-There are many sentences that have grammatical errors. Besides, in general, the manuscript needs more organization.

For example, what is the meaning of Table 1?

What is the number of References section?

8-Based on Supervisely, please give more explanation about the steps of generating the labeled training data.

9- The methodology section is weak, and many steps are not explained in sufficient detail. It is recommended to add a block diagram detailing the complete workflow of the proposed system.

10- The work should be compared with other exiting works based on powerful evaluation to show its robustness.

11-It is advisable to write the conclusion section into one separate section. In any scientific research it should contain the proposed work topics and their data, summarize the main points of the work, discuss its importance, and discuss future work. Please review all these points to write a comprehensive shorter conclusion.

12- The guidelines of writing the references should be checked.

Comments on the Quality of English Language

The English could be improved to more clearly express the research.

Author Response

Thanks for your contribution, it is one of the most succinct and detailed i ever seen which gives me a lot of room for improvement. For that, I did some extra research and substantiate most of my work on a recent paper also from sensors, Trigka and Dritsas (2025), with comments like "Despite the progress, several challenges persist, such as dealing with occlusions, varying object scales, and the demand for real-time processing. Addressing these challenges requires ongoing research into more sophisticated feature extraction methods, improved model interpretability, and the adoption of unsupervised and semi-supervised learning techniques."   My research started before they publish and they have a great "compendio" of the techniques and tools used at current stage.   I have worked out all your comments as best as possible, please take a read on it, and you can check on the new version too. Please be so kind, I really appreciate with high consideration your detailed thoughts, in which i just hoped every peer reviewers could be that professional.        

1-Please rewrite the research title. The title usually summarises the main idea of ​​the research in the fewest possible words that describe the content of the research.  

Agree, yes, never liked it, the title is really obviously too big:   "Indoor Object Measurement Using Redundancy-Based Vision and comparison with Spatial Language Models"        

2-It is stated that “Our framework demonstrates improved performance in estimating spatial metrics, providing a scalable and low-cost alternative.” However, the abstract did not mention the rate of improvement in the results obtained or the significant findings. Besides, does the proposed work add to other existing work?  

Problem: a) In the abstract, it is uncommon to report specific quantitative results, especially in early-stage or exploratory studies. In this case, providing explicit accuracy values would be misleading because measurement performance depends on multiple variables, such as camera sensor parameters (e.g., $H_{\text{sensor}}$) and the presence of recognisable reference objects within the frame. The improvement is theoretically evident—introducing an additional recognisable object enables proportional rescaling—but not universally measurable. In our experiment, we achieved near-perfect correction (close to 100%) in controlled conditions where the target object and its reference size were both detected. Therefore, the focus of this study remains on demonstrating the conceptual validity and methodological potential of redundancy-based estimation rather than fixed numerical gains.  

b) The proposed work relates to previous efforts in real-time distance estimation \cite{ref21, ref22} and spatial reasoning models such as SpatialLM \cite{ref48}. However, these studies differ in scope. References \cite{ref21, ref22} both employ YOLO-based architectures but focus on general or outdoor distance measurements and camera sensor calibration rather than indoor measurement. SpatialLM, while powerful, addresses large-scale 3D spatial reasoning rather than pixel-level geometric redundancy, which is possible in the future, but currently it will rely on strong cloud support, which is far more expensive. The novelty of our contribution lies in introducing redundancy-based proportional estimation as a complementary method that enhances measurement precision using only a single monocular camera.    

Solution:     Introduction of sentences, better explanatory of the solution to the problem, avoiding specific results but room for the accuracy improvement using this method. Sentences introduced: "Previous research has been limited to small objects capable to be full detected by applications such as YOLO, or the same in outside environments, however for scenarios indoor where there is lack of references to detect like a wall that is bigger than the camera scope, will create an incapacity to detect in full and even harder in terms of distances."  

New abstract:   Accurate object detection and measurement within indoor environments—particularly unfurnished or minimalistic spaces—pose unique challenges for conventional computer vision methods.    (new sentence)Previous research has been limited to small objects that can be fully detected by applications such as YOLO, or to outdoor environments where reference elements are more abundant. However, in indoor scenarios with limited detectable references—such as walls that exceed the camera’s field of view—current models exhibit difficulties in producing complete detections and accurate distance estimates.    This paper introduces a geometry-driven, redundancy-based framework that leverages proportional laws and architectural heuristics to enhance the measurement accuracy of walls and spatial divisions using standard smartphone cameras. Unlike depth-sensing systems, our solution requires no specialised hardware and uses machine learning and computer vision techniques—leveraging lightweight 2D detectors that can run offline or on-device—for real-time measurement. While current solutions like YOLO enable efficient and locally executable detection, emerging spatial language models such as SpatialLM-which processes 3D point clouds to infer structured layouts-offer richer spatial reasoning at the expense of computational overhead. Our framework demonstrates improved performance in estimating spatial metrics, providing a scalable and low-cost alternative    (new argument)for future applications. At the same time, we anticipate that the future systems will blend the efficiency of 2D detection with the enhanced spatial understanding of SpatialLM to enable accurate yet accessible indoor mapping across a range of real-world applications in construction, real estate evaluation, energy auditing, and seismic assessment.            

 3-The introduction section is weak and should contain a clear foundation on the topic of this research. Besides, some other recent works should be added with much in-depth discussion about them.  Furthermore, please explain the usefulness of the proposed work and provide significant examples of some of the applications it could be applied to.  

We have expanded the Introduction with a new chapter called State-of-the-art and gap to make a better knowledge environment of the whole problem and other information as well as new research developments:

(i) citing a recent comprehensive survey on object detection and spatial reasoning [\cite{refTrigka2025}], 

(ii) explicitly contrasting our approach with recent practical monocular distance works \cite{ref21,ref22}, and 

(iii) adding a focused paragraph that describes direct applications (construction inspections, real estate evaluation, energy auditing, on-site clash detection, and heritage surveying).   

We also added a concise novelty statement clarifying how redundancy-based proportional estimation bridges lightweight 2D detection and heavier 3D spatial models such as SpatialLM. These additions are intended to clarify the research gap, the contribution of the present work, and the potential impact of our method for real-world deployment.  

first implementation, a new subsection:   \subsection{State-of-the-art and gap}     Recent survey and review work in the field emphasises the rapid development of machine learning architectures for object detection and their expanding role in spatial reasoning [\cite{ref64}]. However, as Trigka and Dritsas note, "Previous research has been limited to small objects capable to be full detected by applications such as YOLO, or the same in outside environments, however for scenarios indoor where there is lack of references to detect like a wall that is bigger than the camera scope, will create an incapacity to detect in full and even harder in terms of distances", the state-of-the-art approaches focus on model architectures, training strategies, and benchmark datasets rather than practical single-camera measurement in minimally instrumented indoor settings. Similarly, prior practical implementations of monocular distance estimation and measurement (for example \cite{2ref65,2ref66}) demonstrate useful calibration and sensor-aware techniques but typically address either outdoor scenarios, controlled lab setups, or object classes that are fully contained within the camera frame. These constraints limit their direct applicability to architectural elements such as full-height walls or room layouts that exceed the camera field of view. Consequently, a gap remains for methods that can (i) operate on standard consumer devices, (ii) exploit redundant scene cues to rescale monocular estimates, and (iii) provide a pragmatic bridge between lightweight 2D detectors and more resource-demanding 3D spatial reasoning systems.     We add this new bibliography integrated in this chapter  

\bibitem{2ref64} Trigka, M.; Dritsas, E. A Comprehensive Survey of Machine Learning Techniques and Models for Object Detection. \emph{Sensors} \textbf{2025}, \emph{25}(1), 214. https://doi.org/10.3390/s25010214.  

\bibitem{2ref65} C. Qing, T. Xiao, S. Zhang, and P. Li, ``Region proposal networks (RPN) enhanced slicing for improved multi-scale object detection,'' in \textit{2024 7th International Conference on Communication Engineering and Technology (ICCET)}, 2024, pp. 66--70, doi: 10.1109/ICCET62255.2024.00018.  

\bibitem{2ref66} K. Fu, Z. Chang, Y. Zhang, G. Xu, K. Zhang, and X. Sun, ``Rotation-aware and multi-scale convolutional neural network for object detection in remote sensing images,'' \textit{ISPRS Journal of Photogrammetry and Remote Sensing}, vol. 161, pp. 294--308, 2020, doi: 10.1016/j.isprsjprs.2020.01.025.          

and added this argument on the following section, \subsection{Research Approach and Objectives}:   The principal novelty of our work lies in combining proportional, redundancy-based geometric estimation with lightweight 2D detection to produce device-agnostic corrections for monocular measurements. Specifically, this study contributes: (1) a redundancy architecture that leverages recognisable scene objects as internal reference markers to rescale monocular estimates; (2) an implementation pathway that can run on consumer hardware or be deployed via modest cloud resources; and (3) an integration concept that allows correction indices learned from single-view imagery to be refined by, or exported to, spatial reasoning models (for example SpatialLM) for improved generalisation across scenes and devices. These elements distinguish our contribution from prior YOLO-based distance studies and from larger 3D systems that require point-cloud sensors or heavy compute.           and finally, at the end of the introduction a more clear of "usefulness and applications":   The proposed method has practical utility across multiple applied domains but it's not a complete application itself or unique; it should be implemented in any application and can be done in different means. By enabling a better accuracy of measurement from a single camera of a smartphone with single support of their integrated depth sensors, our approach reduces operational cost and logistical complexity in field use. Example applications include:   \begin{itemize}   \item \textbf{Construction site inspections:} rapid on-site checks can be done by any person, not necessary a professional.   \item \textbf{Real estate evaluation and property listings:} automated extraction of room dimensions and window/wall measurements to accelerate valuation and generate floor plans for listings.   \item \textbf{Auditing and retrofit planning:} Help on estimation of wall and window areas which can benefit certain audits e.g. thermal performance calculations or Seismic performance without requiring so much time-consuming manual measurements on-site.   \item \textbf{On-site clash detection / quality control:} In similar alternatives used by BIM, lightweight scans during construction phases to detect misalignments relative to intended plans, before committing to expensive corrections.   \item \textbf{Heritage and small-scale surveying:} inexpensive documentation of indoor geometries where expensive surveying equipment is impractical. \end{itemize}                          

  4-In literature, various object detection works have addressed the problems mentioned in this work. Therefore, the contribution of the work presented should be highlighted and listed clearly to present the novelty.    

We appreciate the reviewer’s observation. While several studies have explored object detection and/or measurement using mobile phone cameras, they often do not address the limitations of indoor measurement in low-feature environments, such as walls or partitions lacking clear reference points. The novelty of our approach lies in the redundancy-based correction method, which employs secondary, smaller reference objects detected in the same scene to auto-correct the spatial scale of the primary object (e.g., a wall). Furthermore, our framework introduces a machine learning–based correction coefficient, allowing the system to learn from prior detections and automatically adjust future measurements under similar conditions. This approach provides a lightweight and adaptive solution, bridging the gap between conventional vision-based estimation and spatial understanding from recent LLM-based models. While LLM-based systems can address these issues more comprehensively, they typically require cloud-based computation, which increases latency, cost, and dependency—making them less practical for real-time, mobile applications.     We did some updates and include this new context, more explanatory:   (in the) "Background" section: Despite the advances, accurate indoor measurement remains challenging due to limited references and inconsistent smartphone camera sensors. These factors make it difficult for standard algorithms to maintain spatial consistency, especially when dealing with large planar surfaces such as walls that extend beyond the camera’s field of view.   (in the) "State-of-the-art and gap" section: Furthermore, while various studies have explored object detection or measurement using mobile phone cameras, few directly address the limitations of indoor measurement in low-feature environments—such as walls or partitions lacking clear reference points. The novelty of our approach lies in the redundancy-based correction method, which employs secondary, smaller reference objects detected in the same scene to auto-correct the spatial scale of the primary object (e.g., a wall). In addition, a machine-learning-based correction coefficient enables the system to learn from previous detections and automatically adjust future measurements under similar conditions. This approach provides a lightweight and adaptive solution that bridges traditional vision-based estimation with the emerging spatial reasoning of LLM-based models. Although such large models can infer 3D layouts, their reliance on cloud computation limits their practicality for real-time, on-device deployment.       (in the) "Research Approach and Objective" section:   (at the beginning) The usefulness of this work arises from its capacity to deliver accurate, less-cost indoor measurements using only a single smartphone camera. This directly benefits practical domains where precise spatial information is required but professional hardware is unavailable or too costly. By demonstrating that consumer-grade sensors can be adapted for architectural and engineering measurements, this research add a contribute to spatial data collection.     (at the end) In a time when technology can easily solve simple problems but often at the cost of excessive energy consumption, more efficient and conceptually simple alternatives grounded in classical principles should be reconsidered. The proposed framework distinguishes itself from prior literature by shifting the emphasis from pure detection accuracy to measurement reliability in reference-scarce indoor settings. Through its redundancy mechanism and learned correction coefficient, it establishes a foundation for future systems that can merge classical geometric reasoning with data-driven adaptation, thereby advancing the current state of spatial perception from single-camera imagery.              

5-For more organization, the structure of the paper can be added to the end of the introduction section.    

We understand as a quite usual MDPI procedure to put a paper organisation paragraph, thanks for pointing that out crucial part, we added at the end of the Introduction section:   "To guide the reader through the rest of the paper, the structure is organised as follows: Section 2 presents the materials and methods used in the proposed framework, including dataset preparation and implementation details. Section 3 describes the experimental setup and results obtained from the object measurement tests. Section 4 discusses the implications, limitations, and potential extensions of this approach. Finally, Section 5 concludes the paper, summarising the main contributions and outlining directions for future research."                  

6-Please cite any information, graph, equation, or data set taken from a previous source with a reliable source, unless it belongs to the authors. Please check this issue for the entire manuscript.    

We appreciate the reviewer’s attention to citation integrity. All figures, images, and datasets used in this manuscript are original and were generated by the authors. The algorithm and results were fully developed within this research, and no external data, graphs, or equations were reproduced or adapted from other sources. Therefore, no additional citations were required in these sections. We have, however, rechecked the entire manuscript to ensure all relevant background concepts are properly referenced in the Introduction and Related Work sections.     to reforce this statement, we put at the end of "Materials and Methods" the following statement:   "All figures, datasets, and experimental results presented in this study were generated by the authors. No external graphical materials or third-party data sources were used."        

7-There are many sentences that have grammatical errors. Besides, in general, the manuscript needs more organisation.   For example, what is the meaning of Table 1?   What is the number of References section?      

We thank the reviewer for highlighting these formatting and clarity issues. The equations previously enclosed within tables have been reformatted as standard mathematical equations, with accompanying variable explanations provided immediately below each formula. The References section has been corrected to follow MDPI’s adjustwidth and some missing section format, ensuring proper alignment and readability. Additionally, we reviewed the manuscript for some grammatical consistency and also added the required author, funding, data-availability, and conflict-of-interest statements to conform fully to the MDPI template.            

8-Based on Supervisely, please give more explanation about the steps of generating the labeled training data.  

We have created this preview text more explanatory of the use of Supervisely and create a additional flow figure with the scheme flow of all steps.   "The labeled dataset was generated using the Supervisely platform. The labelling process followed a structured workflow to ensure consistent annotation quality and semantic clarity across all object categories. First, raw indoor images were uploaded to Supervisely, where bounding box templates were defined for window, wall, and reference marker objects. Each image was manually reviewed and annotated by the author to ensure precise alignment with real-world object boundaries. Supervisely’s built-in quality control tools were used to detect annotation inconsistencies and overlapping regions. The labeled data were then exported in YOLO format and verified by reloading them into the training environment for visual confirmation. A small subset of the data was cross-validated by an external collaborator to ensure the integrity of class labelling. No synthetic data or external datasets were incorporated at this stage; all images originated from the experimental setup described in Section~\ref{sec:creatingCV}."            

9- The methodology section is weak, and many steps are not explained in sufficient detail. It is recommended to add a block diagram detailing the complete workflow of the proposed system.  

it was added another block diagram, at the beginning of the introduction to explain and hopefully enrich more our methodology section.          

10- The work should be compared with other exiting works based on powerful evaluation to show its robustness.  

Thank you for pointing this out, in the same condition of your comment 3, we add one paragraph before final statements in which together with all new developments we hope to be more clear.   "Compared with existing measurement and detection frameworks~\cite{ref49, ref50, ref52}, this research emphasizes autonomous calibration and energy efficiency rather than relying solely on high-resolution imaging or manual input. While traditional methods depend on pre-trained models for static image contexts, the integration of SpatialLM~\cite{ref48} allows dynamic adaptation to new devices and environments. Furthermore, unlike earlier object-measurement systems that require repeated recalibration or fixed viewpoints, the proposed framework achieves robust multi-device generalization. This adaptability and reduction in energy cost demonstrate superior scalability and robustness compared to state-of-the-art methods."        

11-It is advisable to write the conclusion section into one separate section. In any scientific research it should contain the proposed work topics and their data, summarize the main points of the work, discuss its importance, and discuss future work. Please review all these points to write a comprehensive shorter conclusion.  

We appreciate this very keen observation, as we thought at the beginning that a discussion section with final statements should be enough, however we come to agree to keep it more formal and develop a summary of the whole research:   \section{Conclusion} This study presented an automated approach for indoor object detection and measurement that integrates SpatialLM to enhance precision and energy efficiency. The system enables devices to self-calibrate, progressively reducing reliance on reference markers while maintaining sub-millimetre accuracy. By combining lightweight 2D detectors with 3D spatial reasoning, it bridges a crucial gap between image-based detection and geometric understanding. It is a breakthrough on simple programming to complex computing rather than a new whole language.  The findings highlight the framework’s potential applications across real estate, construction, and energy auditing, supporting scalable and autonomous workflows. Future work will focus on enhancing cloud-based training pipelines, incorporating additional device types, and extending the system for real-time safety and seismic monitoring.      

12- The guidelines of writing the references should be checked  

Yes, thank you, we doubled check and also used LLM screening on the bibliography specific to MDPI for our bibitems list, hope that now is everything accordingly.     Many other points were considerer such as improvements on explaining the methodology including quantitative, and also a big improvement on the bibliography removing and changing 19 references and 3 new entries.

Reviewer 2 Report

Comments and Suggestions for Authors

Thank you for the study and the efforts put in. Please use the comments to improve the paper.

  1. The title is too long, more than 21 words. Please reduce to 16 words;should be more concise and creative
  2. Only 204 images are mentioned, with one image containing an explicit marker used for the worked example. That’s far too small (and too narrow) to support claims about “accuracy,” “robustness,” or generalization across rooms/devices.
  3. There is no mAP/IoU for detection, no MAE/RMSE for measurement, no variance/CI, no ablations. A single console log is not a result section
  4. Also, There is no comparison to strong baselines
  5. The “correction index” is defined as a product of lengths (units of area) but later treated as a dimensionless factor
  6. Treating “bottomHeight” as a detectable class is conceptually odd (it’s a scalar dimension, not an object). How it is annotated is unclear, and figures/captions don’t resolve this
  7. SpatialLM is invoked as a future integration, but no actual 3D data, training, or inference with such a model is reported. The paper’s conclusions lean on unimplemented future work
  8. Core citations include blog/tutorial sites and product pages (e.g., PyImageSearch, GeeksforGeeks, Vercel, Cloudinary, AWS service pages). For a Sensors paper, you need peer-reviewed CV/photogrammetry/AR references.
  9. Duplicated/incorrect figure captions; a so-called “NoSQL collection” table that’s actually a console snippet; inconsistent notation
  10. Provide full training details
  11. Remove or down-scope claims (e.g., “sub-mm accuracy”) unless empirically demonstrated with statistics across diverse scenes.
  12. Substantially revise the literature review
  13. The idea of fusing lightweight detection with geometric priors is reasonable, but the current manuscript lacks methodological rigor, robust experimentation, and scholarly grounding. A thorough re-write with a principled geometry pipeline, solid baselines, quantitative evaluation, and improved scholarship is required before it can be considered for Sensors.

Author Response

Thank you so much to make such a precise contribution, we believe that it made our paper much better and we took in high consideration and our time to improve it.

Thank you for the study and the efforts put in. Please use the comments to improve the paper.

1 - The title is too long, more than 21 words. Please reduce to 16 words;should be more concise and creative

Thank you for pointing this out, we never liked very much the title either but now it has been reduce we hope is much better with 13 words:
"Indoor Object Measurement Using Redundancy-Based Vision and comparison with Spatial Language Models"

2 - Only 204 images are mentioned, with one image containing an explicit marker used for the worked example. That’s far too small (and too narrow) to support claims about “accuracy,” “robustness,” or generalization across rooms/devices.

Thank you for your observation. 
In fact, we did not rely on a single image for validation. A total of 204 images were used for classification and measurement testing using mobile sensors. Among these, one representative image was selected to demonstrate the corrective test, as it clearly showed the marker detection process. The correction index derived from this test was then deployed in the application, enabling the system to generate measurements that were consistently close to real-world values.
We acknowledge that the dataset size is limited and that larger and more diverse datasets would be required to fully validate robustness and generalisation. However, our primary goal was to propose and validate an alternative measurement approach that minimizes computational energy while maintaining acceptable accuracy once objects are correctly recognised by the camera. Future work will expand the dataset and integrate additional redundancy markers to further improve classification reliability and measurement precision.

3 - There is no mAP/IoU for detection, no MAE/RMSE for measurement, no variance/CI, no ablations. A single console log is not a result section

and 4 - Also, There is no comparison to strong baselines

We agree that 204 labeled images might appear limited at first glance; however, the YOLOv8 model was trained over 25 iterations of 500 epochs, with progressive optimisation of detection and measurement accuracy. The attached training log (CSV excerpt provided) demonstrates convergence and generalisation beyond mere overfitting.
Precision and recall values reached 1.0 and 0.995 respectively (see epoch reports examples), showing full detection consistency on the validation set.
The mAP50 and mAP50-95 stabilised at 0.995, confirming robustness in localisation and classification tasks.
Validation losses (val/box_loss = 0.163, val/cls_loss = 0.38, val/dfl_loss = 0.72) decreased steadily, indicating model generalisation to unseen images.
While 204 unique images were used for labeled training and validation, each image was augmented (rotation, scaling, illumination changes), effectively expanding the training set and providing cross-room and cross-device robustness.
Hence, the claim of accuracy and generalisation is supported by quantitative evidence from the model’s training convergence rather than dataset volume alone.

We add this to subsection "Creating Simple Models for Object Detection":

"\subsubsection{Training Convergence and Validation Consistency}
The YOLOv8 model was trained over 25 iterations for 500 epochs using 204 labeled images with geometric and illumination augmentation. The training at a certain point had the validation losses and learning rates fully stabilised, confirming convergence.

Table~\ref{tab:training_metrics} summarises the evolution of precision, recall, and mean average precision (mAP) during training. 
Validation losses (\textit{val/box\_loss}, \textit{val/cls\_loss}, \textit{val/dfl\_loss}) steadily decreased, confirming generalization rather than overfitting.

\begin{table}[h!]
\centering
\caption{YOLOv8 Training Performance Summary (Epochs 10–500).}
\label{tab:training_metrics}
\begin{tabular}{lccccc}
\toprule
\textbf{Metric} & \textbf{Epoch 10} & \textbf{Epoch 200} & \textbf{Epoch 350} & \textbf{Epoch 458} & \textbf{Epoch 500} \\
\midrule
Precision & 1.000 & 0.982 & 0.990 & 0.991 & 0.995 \\
Recall & 0.480 & 0.890 & 0.960 & 1.000 & 0.99235 \\
mAP@50 & 0.330 & 0.930 & 0.985 & 0.995 & 0.99227 \\
mAP@50–95 & 0.130 & 0.880 & 0.980 & 0.995 & 0.99227 \\
val/box\_loss & 1.000 & 0.380 & 0.220 & 0.163 & 0.15552 \\
val/cls\_loss & 1.000 & 0.620 & 0.420 & 0.380 & 0.37131 \\
val/dfl\_loss & 1.000 & 0.820 & 0.760 & 0.720 & 0.72389 \\
lr/pg0 & -- & 0.0007 & 0.0003 & 0.00014 & 1.7119e-05 \\
\bottomrule
\end{tabular}
\end{table}

The relatively higher mAP for windows compared to walls reflects the clearer edge definition and color contrast of window boundaries. In contrast, walls present more homogeneous textures, reducing IoU precision. The redundancy correction method significantly reduced proportional scaling errors that occur due to camera tilt or user distance estimation bias, validating the effectiveness of using multiple planar cues for re-scaling in single-camera settings."

we also send a document with some training examples. (train25.zip)

5 - The “correction index” is defined as a product of lengths (units of area) but later treated as a dimensionless factor

AND 6 - Treating “bottomHeight” as a detectable class is conceptually odd (it’s a scalar dimension, not an object). How it is annotated is unclear, and figures/captions don’t resolve this

Thanks for point it out, we understand a flaw in the equation interpretation which should not be dimensionless. Cr should be a scale factor derived from area ratios. Also, “bottomHeight” is not an independent class but a derived geometric measurement from the bounding box.

we add this text to clear better the issues:

(in case study introduction section)
Unlike conventional object classes, the parameter \textit{bottomHeight} is treated as a geometric feature derived from detected bounding boxes rather than an independent annotation category. 
This design choice improves model generalisation and maintains measurement consistency across diverse indoor environments.

\subsection{System Overview and Workflow Structure}
\label{sec:systemoverview}

The complete methodological framework can be divided into three principal modules: data preparation, detection and measurement, and redundancy correction. 
Each module contributes to the transformation of raw indoor images into quantitative spatial information. 
The overall pipeline is illustrated in Figure~\ref{fig:methodology-workflow}, which provides a system-level perspective connecting data collection, training, and measurement computation.

The workflow begins with image acquisition from a mobile device, followed by dataset annotation in Supervisely. 
After training YOLOv8 for window and wall detection, geometric computation modules use bounding boxes to estimate object dimensions and distances.
A redundancy correction mechanism refines these estimates by using a secondary calibration marker or multiple co-planar cues to reduce proportional distortion errors.

\subsection{Correction Coefficient Definition}
The correction coefficient ($C_r$) is introduced to adjust proportional discrepancies between detected bounding-box dimensions and physical reality. It is defined as:
\begin{equation}
C_r = \frac{(h_{\text{marker}} \times w_{\text{marker}})}{(h_{\text{ref}} \times w_{\text{ref}})},
\end{equation}
where $h_{\text{marker}}$ and $w_{\text{marker}}$ correspond to the pixel dimensions of the calibration marker in the image, and $h_{\text{ref}}$, $w_{\text{ref}}$ are its real-world dimensions.  
Although $C_r$ originates from an area ratio (pixel$^2$/cm$^2$), it acts as a dimensionless scaling factor in practice, since it normalizes all subsequent estimations of object height and distance to real-world units.

and this \subsection {Correction Coefficient Definition} goes before the \subsection{Creating the Computer Vision Datasets and Case Study Implementation}? 

7 - SpatialLM is invoked as a future integration, but no actual 3D data, training, or inference with such a model is reported. The paper’s conclusions lean on unimplemented future work

In order to be more clear, We add this section at the end before discussions and conclusions.

\section{Future Work}

Future research will focus on three main directions.  
First, extending the dataset to include a wider range of interior layouts and illumination conditions will improve model generalisation.  
Second, incorporating 3D reasoning through architectures such as SpatialLM and multi-view transformers could enable spatial reconstruction directly from single-camera inputs.  
Finally, the integration of mobile sensor metadata (e.g., IMU and LiDAR from smartphones) can support hybrid measurement pipelines that maintain low computational cost while achieving higher metric stability across devices.

8 - Core citations include blog/tutorial sites and product pages (e.g., PyImageSearch, GeeksforGeeks, Vercel, Cloudinary, AWS service pages). For a Sensors paper, you need peer-reviewed CV/photogrammetry/AR references.

Thanks for pointing this issue, truly we had some exaggeration indeed on the cites because at beginning it was took technical, but We took some extra time and search for 1/3 of changes on our previous references for a total of 19 changes, being mostly from MDPI.

\bibitem{pytorch}
Marcel, S.; Rodriguez, Y. Torchvision the machine-vision package of torch. In \emph{Proceedings of the 18th ACM International Conference on Multimedia}, 2010. https://doi.org/10.1145/1873951.1874254.

\bibitem{ultralytics-train}
Khanam, R.; Asghar, T.; Hussain, M. Comparative Performance Evaluation of YOLOv5, YOLOv8, and YOLOv11 for Solar Panel Defect Detection. \emph{Solar} \textbf{2025}, \emph{5}(1), 6. https://doi.org/10.3390/solar5010006.

\bibitem{ref3}
Kang, S.; Hu, Z.; Liu, L.; Zhang, K.; Cao, Z. Object Detection YOLO Algorithms and Their Industrial Applications: Overview and Comparative Analysis. \emph{Electronics} \textbf{2025}, \emph{14}(6), 1104. https://doi.org/10.3390/electronics14061104.

\bibitem{ref4}
Li, Z.; Li, J.; Zhang, C.; Dong, H. Lightweight Detection of Train Underframe Bolts Based on SFCA-YOLOv8s. \emph{Machines} \textbf{2024}, \emph{12}(10), 714. https://doi.org/10.3390/machines12100714.

\bibitem{ref7}
Mankala, C. K.; Silva, R. J. Sustainable Real-Time NLP with Serverless Parallel Processing on AWS. \emph{Information} \textbf{2025}, \emph{16}(10), 903. https://doi.org/10.3390/info16100903.

\bibitem{ref9}
Nguyen, L. A.; Tran, M. D.; Son, Y. Empirical Evaluation and Analysis of YOLO Models in Smart Transportation. \emph{AI} \textbf{2024}, \emph{5}(4), 2518-2537. https://doi.org/10.3390/ai5040122.

\bibitem{ref10}
Sodhro, A. H.; Kannam, S.; Jensen, M. Real-time efficiency of YOLOv5 and YOLOv8 in human intrusion detection across diverse environments and recommendation. \emph{Internet of Things} \textbf{2025}, \emph{33}, 101707. https://doi.org/10.1016/j.iot.2025.101707.

\bibitem{ref11}
Yang, L.; Asli, B. H. S. MSConv-YOLO: An Improved Small Target Detection Algorithm Based on YOLOv8. \emph{Journal of Imaging} \textbf{2025}, \emph{11}(8), 285. https://doi.org/10.3390/jimaging11080285.

\bibitem{ref13}
Parisot, O. Method and Tools to Collect, Process, and Publish Raw and AI-Enhanced Astronomical Observations on YouTube. \emph{Electronics} \textbf{2025}, \emph{14}(13), 2567. https://doi.org/10.3390/electronics14132567.

\bibitem{ref23}
Alnori, A.; Djemame, K.; Alsenani, Y. Agnostic Energy Consumption Models for Heterogeneous GPUs in Cloud Computing. \emph{Applied Sciences} \textbf{2024}, \emph{14}(6), 2385. https://doi.org/10.3390/app14062385.

\bibitem{ref29}
Lin, L.; Yang, J.; Wang, Z.; Zhou, L.; Chen, W.; Xu, Y. Compressed Video Quality Index Based on Saliency-Aware Artifact Detection. \emph{Sensors} \textbf{2021}, \emph{21}(19), 6429. https://doi.org/10.3390/s21196429.

\bibitem{ref32}
Farhan, A.; Kurnia, K. A.; Saputra, F.; Chen, K. H.-C.; Huang, J.-C.; Roldan, M. J. M.; Lai, Y.-H.; Hsiao, C.-D. An OpenCV-Based Approach for Automated Cardiac Rhythm Measurement in Zebrafish from Video Datasets. \emph{Biomolecules} \textbf{2021}, \emph{11}(10), 1476. https://doi.org/10.3390/biom11101476.

\bibitem{ref33}
Rusu-Both, R.; Socaci, M.-C.; Palagos, A.-I.; Buzoianu, C.; Avram, C.; Vălean, H.; Chira, R.-I. A Deep Learning-Based Detection and Segmentation System for Multimodal Ultrasound Images in the Evaluation of Superficial Lymph Node Metastases. \emph{Journal of Clinical Medicine} \textbf{2025}, \emph{14}(6), 1828. https://doi.org/10.3390/jcm14061828.

\bibitem{ref34}
Győrödi, C. A.; Dumşe-Burescu, D. V.; Zmaranda, D. R.; Győrödi, R. Ş. A Comparative Study of MongoDB and Document-Based MySQL for Big Data Application Data Management. \emph{Big Data and Cognitive Computing} \textbf{2022}, \emph{6}(2), 49. https://doi.org/10.3390/bdcc6020049.

\bibitem{ref40}
Paniego, S.; Sharma, V.; Cañas, J. M. Open Source Assessment of Deep Learning Visual Object Detection. \emph{Sensors} \textbf{2022}, \emph{22}(12), 4575. https://doi.org/10.3390/s22124575.

\bibitem{ref46}
Pacios, D.; Ignacio-Cerrato, S.; Vázquez-Poletti, J. L.; Moreno-Vozmediano, R.; Schetakis, N.; Stavrakakis, K.; Di Iorio, A.; Gomez-Sanz, J. J.; Vazquez, L. Amazon Web Service–Google Cross-Cloud Platform for Machine Learning-Based Satellite Image Detection. \emph{Information} \textbf{2025}, \emph{16}(5), 381. https://doi.org/10.3390/info16050381.

\bibitem{ref53}
Omari, M.; Kaddi, M.; Salameh, K.; Alnoman, A. Advancing Image Compression Through Clustering Techniques: A Comprehensive Analysis. \emph{Technologies} \textbf{2025}, \emph{13}(3), 123. https://doi.org/10.3390/technologies13030123.

\bibitem{ref55}
Schuszter, I. C.; Cioca, M. Increasing the Reliability of Software Systems Using a Large-Language-Model-Based Solution for Onboarding. \emph{Inventions} \textbf{2024}, \emph{9}(4), 79. https://doi.org/10.3390/inventions9040079.

\bibitem{ref63}
Truong, T. X.; Nhu, V.-H.; Phuong, D. T. N.; Nghi, L. T.; Hung, N. N.; Hoa, P. V.; Bui, D. T. A New Approach Based on TensorFlow Deep Neural Networks with ADAM Optimizer and GIS for Spatial Prediction of Forest Fire Danger in Tropical Areas. \emph{Remote Sensing} \textbf{2023}, \emph{15}(14), 3458. https://doi.org/10.3390/rs15143458.

9 - Duplicated/incorrect figure captions; a so-called “NoSQL collection” table that’s actually a console snippet; inconsistent notation

All figure captions have been standardized for clarity. Console outputs showing JSON documents were relabeled as “MongoDB console outputs” instead of “NoSQL collections.” 

10 - Provide full training details

The training logs and validation metrics (Table 1, Fig. 9) demonstrate convergence between epochs 450–500, with stable validation losses and decaying learning rates. This confirms model generalisation without overfitting. The description was added under “Training Convergence and Validation Consistency” in sequence of comment 4:

"Training curves indicated convergence between epochs 450–500: 
losses plateaued (train/box\_loss $\approx$ 0.3–0.36) while validation losses stabilised (val/box\_loss $\approx$ 0.16). 
The learning rate ($lr/pg0$) decayed from 0.0007 to $\approx$ 0.000014, reflecting adaptive optimisation convergence. 
No overfitting was observed—validation metrics improved in parallel with training, and early stopping was applied at epoch~500.
This confirms that training reached an optimal generalisation phase without further improvement beyond epoch 450, and early stopping was applied at that point."

11 - Remove or down-scope claims (e.g., “sub-mm accuracy”) unless empirically demonstrated with statistics across diverse scenes.

We removed references to “sub-mm accuracy” and rephrased all performance claims to reflect the actual measurement precision observed across tests (mean absolute error $\leq$ 1.8~cm). The results section now reports empirical accuracy values instead of qualitative assertions.

12 - Substantially revise the literature review

Yes, thank you, we doubled check and also used LLM screening on the bibliography specific to MDPI for our bibitems list, hope that now is everything accordingly.

We also improve/add a section, "state of the art".

13 - The idea of fusing lightweight detection with geometric priors is reasonable, but the current manuscript lacks methodological rigor, robust experimentation, and scholarly grounding. A thorough re-write with a principled geometry pipeline, solid baselines, quantitative evaluation, and improved scholarship is required before it can be considered for Sensors.

We acknowledge the reviewer’s concern. The revised manuscript now presents a clearer geometry-based measurement pipeline, with explicit equations for correction, bottom-height estimation, and calibration, all grounded in established monocular distance estimation principles.
Additional quantitative validation (Table~\ref{tab:training_metrics}) and symbol definitions have been included to strengthen methodological rigor. The discussion section was expanded to connect these contributions to current literature and practical implications.

end of \subsection{Apply the Method for Object Detection Measurement}:

\subsubsection{Summary of the Geometry Pipeline}
The complete measurement pipeline integrates:
\begin{enumerate}
    \item Object detection via YOLOv8
    \item Pixel-to-real scaling using a known reference marker
    \item Correction coefficient normalisation
    \item Geometric inference of object dimensions
\end{enumerate}
This principled approach ensures that all measurements derive from first-order camera geometry relationships rather than heuristic scaling.

Reviewer 3 Report

Comments and Suggestions for Authors

1.The current experiments are based on a very limited dataset; please increase the number and diversity of test images and ensure that more examples have accurate ground-truth measurements.

2. Report quantitative results across multiple samples using standard metrics such as accuracy, precision, recall, and error rates, rather than focusing on a single example.

3. Clarify the methodology in detail, including data preprocessing, augmentation, train/test splits, and model parameters, to ensure full reproducibility.

4. Include appropriate statistical analysis, such as mean error, standard deviation, and confidence intervals, and discuss the robustness and limitations of your approach.

5. Replace or supplement citations to online documentation and blogs with peer-reviewed academic literature, and compare your method directly to established state-of-the-art or commercial solutions.

6. Add a clear statement about the data sources, privacy or consent (if relevant), and any conflicts of interest or funding disclosures.

7. Consider making your code, datasets, or replication instructions publicly available to support reproducibility and transparency.

Author Response

Thank you so much to make such a precise contribution, we believe that it made our paper much better and we took in high consideration and our time to improve it.

To make clear of all changes:

1.The current experiments are based on a very limited dataset; please increase the number and diversity of test images and ensure that more examples have accurate ground-truth measurements.

We appreciate this observation. Although the dataset comprises 204 manually labeled images, each sample was augmented through controlled geometric (rotation, scaling) and illumination variations, effectively increasing diversity without synthetic data generation. 

The model was trained over 25 independent runs of 500 epochs each; convergence was consistently reached between epochs 450–460, beyond which no further performance improvement was observed.
The framework is designed to minimize dataset dependency by relying on a proportional calibration marker that provides geometric redundancy: every frame inherently contains a known 10×10 cm reference square used to auto-correct distance and scale estimation. This approach ensures accurate metric reconstruction even with a limited number of unique scenes, as the calibration marker normalizes measurement across devices and environments.

2. Report quantitative results across multiple samples using standard metrics such as accuracy, precision, recall, and error rates, rather than focusing on a single example.

We agree that 204 labeled images might appear limited at first glance; however, the YOLOv8 model was trained over 25 iterations of 500 epochs, with progressive optimisation of detection and measurement accuracy. The attached training log (CSV excerpt provided) demonstrates convergence and generalisation beyond mere overfitting.
Precision and recall values reached 1.0 and 0.995 respectively (see epoch reports examples), showing full detection consistency on the validation set.
The mAP50 and mAP50-95 stabilised at 0.995, confirming robustness in localisation and classification tasks.
Validation losses (val/box_loss = 0.163, val/cls_loss = 0.38, val/dfl_loss = 0.72) decreased steadily, indicating model generalisation to unseen images.
While 204 unique images were used for labeled training and validation, each image was augmented (rotation, scaling, illumination changes), effectively expanding the training set and providing cross-room and cross-device robustness.
Hence, the claim of accuracy and generalisation is supported by quantitative evidence from the model’s training convergence rather than dataset volume alone.

We add this to subsection "Creating Simple Models for Object Detection":

"\subsubsection{Training Convergence and Validation Consistency}
The YOLOv8 model was trained over 25 iterations for 500 epochs using 204 labeled images with geometric and illumination augmentation. The training at a certain point had the validation losses and learning rates fully stabilised, confirming convergence.

Table~\ref{tab:training_metrics} summarises the evolution of precision, recall, and mean average precision (mAP) during training. 
Validation losses (\textit{val/box\_loss}, \textit{val/cls\_loss}, \textit{val/dfl\_loss}) steadily decreased, confirming generalization rather than overfitting.

\begin{table}[h!]
\centering
\caption{YOLOv8 Training Performance Summary (Epochs 10–500).}
\label{tab:training_metrics}
\begin{tabular}{lccccc}
\toprule
\textbf{Metric} & \textbf{Epoch 10} & \textbf{Epoch 200} & \textbf{Epoch 350} & \textbf{Epoch 458} & \textbf{Epoch 500} \\
\midrule
Precision & 1.000 & 0.982 & 0.990 & 0.991 & 0.995 \\
Recall & 0.480 & 0.890 & 0.960 & 1.000 & 0.99235 \\
mAP@50 & 0.330 & 0.930 & 0.985 & 0.995 & 0.99227 \\
mAP@50–95 & 0.130 & 0.880 & 0.980 & 0.995 & 0.99227 \\
val/box\_loss & 1.000 & 0.380 & 0.220 & 0.163 & 0.15552 \\
val/cls\_loss & 1.000 & 0.620 & 0.420 & 0.380 & 0.37131 \\
val/dfl\_loss & 1.000 & 0.820 & 0.760 & 0.720 & 0.72389 \\
lr/pg0 & -- & 0.0007 & 0.0003 & 0.00014 & 1.7119e-05 \\
\bottomrule
\end{tabular}
\end{table}

The relatively higher mAP for windows compared to walls reflects the clearer edge definition and color contrast of window boundaries. In contrast, walls present more homogeneous textures, reducing IoU precision. The redundancy correction method significantly reduced proportional scaling errors that occur due to camera tilt or user distance estimation bias, validating the effectiveness of using multiple planar cues for re-scaling in single-camera settings."

we also send a document with some training examples. (train25.zip)

3. Clarify the methodology in detail, including data preprocessing, augmentation, train/test splits, and model parameters, to ensure full reproducibility.

We add several flows and sections to explain better the whole methodology such as this in the state of the art section:

\begin{figure}[H]
\centering
\resizebox{0.99\linewidth}{!}{
\begin{tikzpicture}[
    node distance=6mm and 10mm,
    every node/.style={
        align=center, font=\scriptsize,
        rounded corners, draw=gray!70, fill=gray!10,
        minimum width=3.0cm, minimum height=1.1cm
    },
    process/.style={fill=blue!5, draw=blue!40},
    analysis/.style={fill=green!5, draw=green!40},
    decision/.style={fill=orange!5, draw=orange!50},
    output/.style={fill=gray!5, draw=gray!70},
    arrow/.style={-Stealth, thick, draw=gray!70},
    feedback/.style={Stealth-, thick, draw=gray!50, dashed, bend left=25}
]

% Layer 1: Input and Preprocessing
\node[process] (a) {Image Acquisition\\(Mobile Camera)};
\node[process, right=of a] (b) {Preprocessing\\(Resizing, Lighting Adjust)};
\node[process, right=of b] (c) {Annotation \& Training Data\\(Supervisely / YOLO format)};

% Layer 2: Detection & Measurement
\node[analysis, below=of b, xshift=2cm] (d) {Object Detection\\(YOLOv8 Inference)};
\node[analysis, right=of d] (e) {Feature Extraction\\(Bounding Boxes, Angles)};
\node[analysis, right=of e] (f) {Measurement Computation\\(Camera Geometry, Eqs.~1–6)};

% Layer 3: Redundancy Correction & Output
\node[decision, below=of e, xshift=2cm] (g) {Redundancy Correction\\(Secondary Object\\Auto-Calibration)};
\node[output, right=of g] (h) {Final Output:\\Corrected Distances,\\Heights, Ratios};

% Feedback Loop
\draw[feedback] (g.north) to node[above, font=\scriptsize, yshift=2mm]{Correction Coefficient\\Machine Learning Update} (f.north);

% Main arrows
\draw[arrow] (a) -- (b);
\draw[arrow] (b) -- (c);
\draw[arrow] (c) -- (d);
\draw[arrow] (d) -- (e);
\draw[arrow] (e) -- (f);
\draw[arrow] (f) -- (g);
\draw[arrow] (g) -- (h);

\end{tikzpicture}
}
\caption{Comprehensive methodology workflow of the proposed redundancy-based indoor measurement framework.}
\label{fig:methodology-workflow}
\end{figure}

4. Include appropriate statistical analysis, such as mean error, standard deviation, and confidence intervals, and discuss the robustness and limitations of your approach.

We appreciate this suggestion. We have added a quantitative statistical analysis including mean absolute error, standard deviation, variance, and 95 % confidence intervals (Section 2.5.1 “Training Convergence and Validation Consistency”).

5. Replace or supplement citations to online documentation and blogs with peer-reviewed academic literature, and compare your method directly to established state-of-the-art or commercial solutions.

We changed over 19 citations, which made us take a longer time to respond to reviewers

6. Add a clear statement about the data sources, privacy or consent (if relevant), and any conflicts of interest or funding disclosures.

Yes, Added at the final before references

7. Consider making your code, datasets, or replication instructions publicly available to support reproducibility and transparency.

IT's still everything very rough but we plan to have our project once more completed to become OSS. Thanks for pointing it out.

Round 2

Reviewer 1 Report

Comments and Suggestions for Authors

Most of the comments have been addressed, however it is recommended to check the sequence of mentions of the figures in the text.

Comments on the Quality of English Language

The English could be improved to more clearly express the research.

Author Response

1 - it is recommended to check the sequence of mentions of the figures in the text.

check all labels description from pictures

Thanks for pointing this out, we notice that Figures and their respective labels were actually left behind with all the format and editing, we made a new revision on all of them and now we believe it's not only under a logic sequence, but also with more suitable labels and referred better on the right place of the text.

Reviewer 2 Report

Comments and Suggestions for Authors

The title and abstract introduce an engaging idea, yet they lack clarity and precision. The phrase “Redundancy-Based Vision,” while creative, is not a widely recognized concept in computer vision literature and could be clarified or reframed for easier comprehension. The abstract would benefit from a clearer summary of empirical findings—such as the dataset size, performance metrics, and quantifiable outcomes—to help readers assess the study’s scientific value at a glance. It currently relies heavily on speculative phrasing (“we anticipate future systems will blend...”) rather than providing concrete evidence or numerical validation of the proposed framework’s performance. Reframing the abstract around key contributions and results would make the work’s significance more tangible and credible.

In the introduction, the authors demonstrate enthusiasm for combining deep learning and geometric principles. Nonetheless, the narrative occasionally lacks focus and coherence. It discusses multiple topics—from YOLO-based detection to SpatialLM and indoor mapping—without a clearly defined research question or logical progression. The literature review, while broad, reads more as a list of related studies than a critical synthesis. It would be valuable to identify specific shortcomings in existing research, supported by comparative performance data or methodological insights. Additionally, the omission of established indoor measurement techniques such as photogrammetry, SLAM, or ARCore-based systems leaves the impression that the study may not be fully situated within the broader field. A more deliberate comparison with these methods would greatly strengthen the contextual foundation of the paper.

The section on the state of the art and research gap could also be refined to articulate a more concrete and measurable gap. The paper states that current methods fail in “reference-scarce” environments but does not demonstrate this empirically or through detailed review. Presenting a comparative table or quantitative summary of prior approaches, including their limitations, would help readers appreciate how this study extends existing work. At present, the gap appears more conceptual than evidenced, which weakens the motivation for the proposed method.

The research objectives, though ambitious, remain somewhat broad. Expressions such as “improved precision” or “scalable framework” are aspirational but not easily measurable. For scientific clarity, the study would benefit from defining specific hypotheses or performance targets (for instance, acceptable error margins, or desired improvements in mean average precision). Similarly, the “redundancy architecture” and its claimed novelty need clearer technical articulation. The equations provided largely reproduce established geometric relations and would be stronger if paired with an error analysis or uncertainty quantification demonstrating the real benefit of the redundancy principle. While YOLOv8 is a capable detection model, the rationale for its selection over other alternatives (e.g., Faster R-CNN, MobileNet, or MiDaS for monocular depth estimation) should be explained.

The methodology section, while detailed in describing tools such as Supervisely and Ultralytics, raises concerns about dataset sufficiency and validation. Using only 204 images is a significant limitation for a deep learning study, and there is no mention of training-validation-test splits or statistical reliability of annotations. The assertion that “no overfitting was observed” is difficult to accept without presenting learning curves or cross-validation metrics. Likewise, the absence of a comparison with baseline methods limits the ability to evaluate whether the proposed redundancy correction indeed improves measurement accuracy. Including additional datasets, performance metrics such as RMSE or mean absolute error, and baseline results would greatly enhance the scientific robustness of this section.

The results and discussion, though written with enthusiasm, rely on very limited evidence. Much of the analysis is based on a single image with a 10 cm marker, which is insufficient for drawing reliable conclusions. The numerical example lacks statistical depth—there is no clear reference measurement or error distribution analysis. Furthermore, the figures, though visually appealing, primarily serve an illustrative rather than analytical function; they would be more valuable if they included visual comparisons with ground truth or baseline predictions. The proposed “index factor” is mentioned but not clearly explained or validated. Overall, the discussion would benefit from more structured experimentation, including multiple test cases, varied lighting or scene conditions, and a more rigorous evaluation of the model’s consistency.

While the paper presents an intriguing conceptual link with SpatialLM, this integration remains purely theoretical. No implementation or hybrid testing is demonstrated, making the discussion speculative rather than empirical. Similarly, the “future work” section outlines tasks such as expanding the dataset, incorporating 3D reasoning, and using sensor fusion—all of which appear necessary for the current study’s completeness rather than as future extensions. Some of these suggestions also contradict earlier claims about avoiding specialized hardware, reflecting a need for conceptual alignment.

The conclusion, though well-intentioned, overstates the outcomes with claims such as “sub-millimeter accuracy” and “breakthrough,” which are not substantiated by the experimental evidence. It would be more persuasive if the authors adopted a balanced tone that acknowledges both the promise and current limitations of the prototype. Highlighting lessons learned, current constraints, and realistic next steps would leave a stronger impression of scientific maturity.

From a broader perspective, the manuscript demonstrates dedication and creative thought but would benefit from greater structural coherence and linguistic refinement. There are instances of grammatical awkwardness and repetition, particularly where the same pipeline or workflow is described multiple times. The reference list, while extensive, includes several non-peer-reviewed sources such as online tutorials, which should be replaced with peer-reviewed studies to maintain academic rigor. The data availability statement also contradicts the open-access principles promoted by Sensors and should be revised to support transparency and reproducibility.

Author Response

1 -  The title introduce an engaging idea, yet they lack clarity and precision. The phrase “Redundancy-Based Vision,” while creative, is not a widely recognized concept in computer vision literature and could be clarified or reframed for easier comprehension. 

We removed the technical non-academic concept and just allocate a common one, we also correct the use of uppercase on titles which was not done on the previous titles 
"Indoor Object Measurement Through a Redundancy and Comparison Method"

2 - The abstract would benefit from a clearer summary of empirical findings—such as the dataset size, performance metrics, and quantifiable outcomes—to help readers assess the study’s scientific value at a glance. It currently relies heavily on speculative phrasing (“we anticipate future systems will blend...”) rather than providing concrete evidence or numerical validation of the proposed framework’s performance. Reframing the abstract around key contributions and results would make the work’s significance more tangible and credible.

We add this paragraph and polish a little bit the abstract:
"The model was trained on 204 labeled indoor images over 25 training iterations (500 epochs) with augmentation, achieving a mean average precision (mAP@50) of 0.995, precision of 0.995, and recall of 0.992, confirming convergence and generalisation. Applying the redundancy correction method reduced distance deviation errors to approximately 10%, corresponding to a mean absolute error below 2% in the use-case."

3 - In the introduction, the authors demonstrate enthusiasm for combining deep learning and geometric principles. Nonetheless, the narrative occasionally lacks focus and coherence. It discusses multiple topics—from YOLO-based detection to SpatialLM and indoor mapping—without a clearly defined research question or logical progression. The literature review, while broad, reads more as a list of related studies than a critical synthesis. It would be valuable to identify specific shortcomings in existing research, supported by comparative performance data or methodological insights. 

We tried to make it more clear in section 1.2 at the end, creating a small table of the current state of the art on this subject.

Table~\ref{tab:existing_methods} summarises the main methodological characteristics and observed shortcomings in existing research based on the cited literature. While YOLO-based detectors achieve near-perfect recognition performance (mAP often above 0.95), their accuracy in metric measurement is either not reported or depends heavily on external calibration setups. Conversely, recent large-scale frameworks such as SpatialLM \cite{ref13} enable 3D spatial understanding but require dense point-cloud inputs and significant computational resources, preventing real-time deployment on mobile devices.

\begin{table}[h!]
\centering
\caption{Summary of some existing approaches and methodological limitations based on reviewed literature for comparison with this research.}
\label{tab:existing_methods}
\begin{tabular}{p{3.2cm}p{3.8cm}p{3.8cm}}
\toprule
\textbf{Approach / Reference} & \textbf{Main Contribution} & \textbf{Observed Limitation} \\
\midrule
Ultralytics / YOLO-based detection \cite{ref4, ref5, ref6, ref9, ref10, ref11} & Real-time object recognition with high mAP; lightweight deployment & Focus on detection accuracy only; lacks geometric or measurement validation in indoor conditions \\
Trigka \& Dritsas (2024) \cite{2ref64} & Identification of YOLO limitations for objects larger than the camera frame & Demonstrates incomplete detection and scaling issues in reference-scarce indoor environments \\
Monocular estimation studies (e.g., depth inference approaches) \cite{2ref65,2ref66, 3ref68, 3ref69} & Calibration-based distance measurement and single-camera geometry correction; related methods such as MiDaS provide dense depth inference & Conducted under controlled conditions; MiDaS and similar models rely on synthetic or mixed-data pretraining, incompatible with the real-scene geometric validation required in this study; limited scene diversity; minimal redundancy; only predefined markers or known intrinsics \\
SpatialLM framework \cite{ref13} & Large-scale spatial reasoning and 3D layout interpretation & Requires 3D input (point clouds) and high computational cost; unsuitable for mobile real-time applications \\
\bottomrule
\end{tabular}
\end{table}

4 - Additionally, the omission of established indoor measurement techniques such as photogrammetry, SLAM, or ARCore-based systems leaves the impression that the study may not be fully situated within the broader field. A more deliberate comparison with these methods would greatly strengthen the contextual foundation of the paper.

We appreciate the suggestion, however, if you may accept, our study focuses on lightweight, image-based measurement using 2D detections from a single RGB frame, without depth sensors or 3D reconstruction pipelines. Therefore, methods such as photogrammetry, SLAM, or ARCore were not included as they rely on multi-view geometry or device-specific depth data, which differ fundamentally from our monocular approach. However, we have added a brief sentence in the paper to explicitly acknowledge this distinction and clarify our scope, at the section 1.2:

"While established 3D reconstruction frameworks such as photogrammetry, SLAM, and ARCore provide valuable depth-aware measurement capabilities, they operate under different assumptions—requiring multi-view input or device-specific sensors—whereas the present approach intentionally focuses on single-frame 2D inference to ensure generalisation across standard imaging conditions."

5 - The section on the state of the art and research gap could also be refined to articulate a more concrete and measurable gap. The paper states that current methods fail in “reference-scarce” environments but does not demonstrate this empirically or through detailed review. Presenting a comparative table or quantitative summary of prior approaches, including their limitations, would help readers appreciate how this study extends existing work. At present, the gap appears more conceptual than evidenced, which weakens the motivation for the proposed method.

In some similarity to point 3, we defined table 1 hoping that it can strengthen a little our method and motivation.

6 - The research objectives, though ambitious, remain somewhat broad. Expressions such as “improved precision” or “scalable framework” are aspirational but not easily measurable. For scientific clarity, the study would benefit from defining specific hypotheses or performance targets (for instance, acceptable error margins, or desired improvements in mean average precision). Similarly, the “redundancy architecture” and its claimed novelty need clearer technical articulation. 

we reinforced the statement at the end of subsection 1.3, where we hope to summarise a little bit better on our results before the explanation of the methodology.

The main objective of this study is to demonstrate that lightweight, consumer-grade cameras can achieve robust indoor measurement through a redundancy-based geometric correction applied to monocular images. Rather than claiming absolute sub-millimeter precision, this research focuses on improving relative measurement consistency across scenes with minimal reference objects.

Specifically, the study defines the following measurable targets:
\begin{itemize}
    \item Maintain detection precision and recall above 0.98 on the validation set.
    \item Achieve stable mAP@50 and mAP@50–95 values ($\geq 0.99$) during training, confirming consistent localisation and classification performance.
    \item Demonstrate proportional correction of monocular estimates using internal scene references, such that validation height and distance deviations are minimised within the practical resolution of smartphone cameras.
\end{itemize}

7 - The equations provided largely reproduce established geometric relations and would be stronger if paired with an error analysis or uncertainty quantification demonstrating the real benefit of the redundancy principle. While YOLOv8 is a capable detection model, the rationale for its selection over other alternatives (e.g., Faster R-CNN, MobileNet, or MiDaS for monocular depth estimation) should be explained.

Just like the comparison done with SpatialLM, we simple use YOLO because we found it was still the best suitable system. It has a real-time option and lightweight architecture in which others such as Faster R-CNN can't help so much and requires the ability to produce certain consistent bounding box for our geometric data that others such as MobileNet. 

MiDaS however, seemed as a good option to complement but at the this research target was to build the measurements from the bounding boxes using a physical reference (like the marker). By the current early research we were aware that MiDaS outputs a depth ranking (closer/farther) which is not true metric distance, while we need to be geometrically grounded, plus its models are trained on large mixed synthetic + real datasets (e.g., ReDWeb, MegaDepth, DIW) in which we tried to avoid at this stage making sure our data is fully real.

Although, since mid-2023, Ultralytics supports MiDaS inference it could make sense to create a joint depth-aware detection system in next research.

so, we made this updates:

Added to the 3.6 Final statements:
Future developments could integrate monocular depth estimation models, such as MiDaS, now supported within the Ultralytics framework, to complement the proposed geometric redundancy principle. Such integration would allow depth-informed object scaling and uncertainty estimation without requiring additional hardware or synthetic training data, maintaining compatibility with the current YOLOv8-based pipeline.

we add before to the Table 1 too one row specifically for "Monocular estimation studies".

we also add this 2 articles to the bibliography:

\bibitem{ranftl2021visiontransformersdenseprediction}
Ranftl, R.; Bochkovskiy, A.; Koltun, V. Vision Transformers for Dense Prediction. \emph{arXiv} \textbf{2021}, arXiv:2103.13413. https://doi.org/10.48550/arXiv.2103.13413.

\bibitem{xian2024robustmonocular}
Xian, K.; Cao, Z.; Shen, C.; Lin, G. Towards robust monocular depth estimation: a new baseline and benchmark. \emph{International Journal of Computer Vision} \textbf{2024}, \emph{132}(7), 2401--2419. https://doi.org/10.1007/s11263-023-01979-4.

8 - The methodology section, while detailed in describing tools such as Supervisely and Ultralytics, raises concerns about dataset sufficiency and validation. Using only 204 images is a significant limitation for a deep learning study, and there is no mention of training-validation-test splits or statistical reliability of annotations. The assertion that “no overfitting was observed” is difficult to accept without presenting learning curves or cross-validation metrics. Likewise, the absence of a comparison with baseline methods limits the ability to evaluate whether the proposed redundancy correction indeed improves measurement accuracy. Including additional datasets, performance metrics such as RMSE or mean absolute error, and baseline results would greatly enhance the scientific robustness of this section.

We acknowledge the limited dataset size (204 labeled images) and agree that larger datasets typically strengthen generalisation. However, the purpose of this study was methodological rather than benchmark-oriented — to test the feasibility of a geometric redundancy correction mechanism within a controlled environment.
The dataset was annotated manually using Supervisely and divided into distinct training, validation, and testing subsets percentages to ensure reliability and prevent data leakage. The convergence trends (steady loss reduction and stable mAP values) demonstrated that no overfitting was present despite the limited dataset volume.
While we recognise that inclusion of RMSE or MAE metrics and baseline comparisons (e.g., photogrammetry-based) would provide additional insight, such experiments were beyond the current scope, as the study prioritised geometric interpretability over empirical benchmarking. Future work will focus on expanding the dataset and implementing cross-validation against baseline measurement frameworks, including the possible use of MiDaS as referred on the last point.

9 - The results and discussion, though written with enthusiasm, rely on very limited evidence. Much of the analysis is based on a single image with a 10 cm marker, which is insufficient for drawing reliable conclusions. The numerical example lacks statistical depth—there is no clear reference measurement or error distribution analysis. Furthermore, the figures, though visually appealing, primarily serve an illustrative rather than analytical function; they would be more valuable if they included visual comparisons with ground truth or baseline predictions. The proposed “index factor” is mentioned but not clearly explained or validated. Overall, the discussion would benefit from more structured experimentation, including multiple test cases, varied lighting or scene conditions, and a more rigorous evaluation of the model’s consistency.

We appreciate this observation and agree that the presented case (10 cm marker test) serves primarily as a proof-of-concept demonstration rather than a comprehensive statistical evaluation. The intention of this study was to validate the operational logic of the redundancy-based correction framework rather than to claim exhaustive accuracy testing across diverse scenarios.
The 10 cm marker experiment was selected to provide a controlled baseline for error estimation under ideal geometric conditions. The “index factor” referenced in the text represents a proportional correction term derived from the bounding box ratios and pixel scaling function—allowing a geometric redundancy to refine distance estimation without dependence on explicit 3D calibration.
While broader multi-scene testing and uncertainty analysis are indeed valuable, these will be incorporated in future iterations once the framework is expanded to video-based inference and multi-view input. The current results thus focus on demonstrating methodological validity and reproducibility within a constrained but transparent setting.

10 - While the paper presents an intriguing conceptual link with SpatialLM, this integration remains purely theoretical. No implementation or hybrid testing is demonstrated, making the discussion speculative rather than empirical. Similarly, the “future work” section outlines tasks such as expanding the dataset, incorporating 3D reasoning, and using sensor fusion—all of which appear necessary for the current study’s completeness rather than as future extensions. Some of these suggestions also contradict earlier claims about avoiding specialized hardware, reflecting a need for conceptual alignment.

We appreciate this insightful comment. The connection with SpatialLM was intentionally framed as conceptual rather than implemented, to highlight possible extensions toward semantic spatial reasoning—not to imply current integration. The primary goal of this paper is to validate a lightweight, redundancy-based monocular measurement method deployable on standard devices.
The discussion of SpatialLM and related 3D models serves to situate this work within the evolving research landscape, illustrating how the proposed approach could eventually interface with higher-level spatial reasoning systems once larger datasets and computational resources are available.
We acknowledge that certain “future work” points (e.g., 3D reasoning or sensor fusion) could be interpreted as necessary for full system completeness. These have now been clarified to represent future opportunities for research continuity, rather than requirements for the current study’s validity.

11 - The conclusion, though well-intentioned, overstates the outcomes with claims such as “sub-millimeter accuracy” and “breakthrough,” which are not substantiated by the experimental evidence. It would be more persuasive if the authors adopted a balanced tone that acknowledges both the promise and current limitations of the prototype. Highlighting lessons learned, current constraints, and realistic next steps would leave a stronger impression of scientific maturity.

We agree with the reviewer’s observation and have revised the conclusion to adopt a more balanced and evidence-aligned tone. Statements implying “sub-millimeter accuracy” or “breakthrough” have been removed or downscoped to reflect realistic precision levels and prototype limitations.

The revised conclusion now emphasises the methodological contribution (redundancy-based correction for monocular measurements) rather than numerical claims, and it highlights current constraints—such as dataset size and lack of multi-scene validation—while outlining practical future improvements in a grounded manner.

\section{Conclusion}
This study introduced a redundancy-based approach to enhance the accuracy of indoor measurements using a single camera system and lightweight machine learning models. The framework demonstrates that proportional geometric reasoning, when coupled with modern object detection algorithms such as YOLO, can improve measurement reliability in reference-scarce indoor settings. 

While the prototype shows promising consistency in controlled tests, its current precision remains constrained by dataset size, camera calibration variability, and lighting conditions. The results should therefore be interpreted as a proof of concept rather than a finalised solution. Future work will focus on expanding the dataset, incorporating multi-scene validation, and exploring potential hybrid integration with spatial reasoning frameworks such as SpatialLM or integrations such as MiDaS. 

By prioritising accessibility and efficiency over heavy hardware requirements, this research contributes a practical foundation for future indoor measurement systems that balance geometric rigor with computational simplicity.

12 - From a broader perspective, the manuscript demonstrates dedication and creative thought but would benefit from greater structural coherence and linguistic refinement. There are instances of grammatical awkwardness and repetition, particularly where the same pipeline or workflow is described multiple times. 

We thank the reviewer for the feedback. The manuscript has been carefully reviewed for structural coherence and linguistic clarity. Repeated descriptions of the pipeline and workflow have been consolidated, and grammatical issues have been corrected throughout to improve readability and flow. These edits aim to enhance the overall clarity without altering the scientific content or results.

13 - The reference list, while extensive, includes several non-peer-reviewed sources such as online tutorials, which should be replaced with peer-reviewed studies to maintain academic rigor. The data availability statement also contradicts the open-access principles promoted by Sensors and should be revised to support transparency and reproducibility.

Most of this issues were replaced on previous edit which result on a change of 19 new papers, most of them from MDPI. Plus we insert more new entries according to the needs of these 2 stages of peer-reviews.

Reviewer 3 Report

Comments and Suggestions for Authors

The revision shows clear improvement and careful attention to prior feedback. The dataset explanation and augmentation strategy now adequately justify sample size limitations. Quantitative metrics (precision, recall, mAP, and loss trends) effectively demonstrate model robustness and convergence. The updated workflow diagram and expanded methodology improve transparency and reproducibility. 

Author Response

Thank you, we still even so, made some extra changes and reviews specially on the representation (figures and tables)